# Towards Efficient Large Language Reasoning Models via Extreme-Ratio Chain-of-Thought Compression

**Yuntian Tang** [* 1] **Bohan Jia** [* 1] **Wenxuan Huang** [* 1] **Lianyue Zhang** [1] **Jiao Xie** [1] **Wenxi Li** [1] **Wei Li** [2] **Jie Hu** [2] **Xinghao Chen** [2] **Rongrong Ji** [3] **Shaohui Lin** [1 4]

## Abstract

Chain-of-Thought (CoT) reasoning successfully enhances the reasoning capabilities of Large Language Models (LLMs), yet it incurs substantial computational overhead for inference. Existing CoT compression methods often suffer from a critical loss of logical fidelity at high compression ratios, resulting in significant performance degradation. To achieve high-fidelity, fast reasoning, we propose a novel EXTreme-RAtio Chain-of-Thought Compression framework, termed Extra-CoT, which aggressively reduces the token budget while preserving answer accuracy. To generate reliable, high-fidelity supervision, we first train a dedicated semantically-preserved compressor on mathematical CoT data with fine-grained annotations. An LLM is then fine-tuned on these compressed pairs via a mixed-ratio supervised fine-tuning (SFT), teaching it to follow a spectrum of compression budgets and providing a stable initialization for reinforcement learning (RL). We further propose Constrained and Hierarchical Ratio Policy Optimization (CHRPO) to explicitly incentivize question-solving ability under lower budgets by a hierarchical reward. Experiments on three mathematical reasoning benchmarks show the superiority of Extra-CoT. For example, on MATH-500 using Qwen3-1.7B, Extra-CoT achieves over 73% token reduction with an accuracy improvement of 0.6%, significantly outperforming state-of-the-art (SOTA) methods. Our code is available at https://github.com/Mwie1024/Extra-CoT.

---

[*]Equal contribution . [1]East China Normal University, China [2]Huawei Foundation Model Dept. [3]Xiamen University, China [4]The Key Laboratory of Advanced Theory and Application in Statistics and Data Science, Ministry of Education, China. Correspondence to: Shaohui Lin <shlin@cs.ecnu.edu.cn>.

*Proceedings of the 43rd International Conference on Machine Learning*, Seoul, South Korea. PMLR 306, 2026. Copyright 2026 by the author(s).

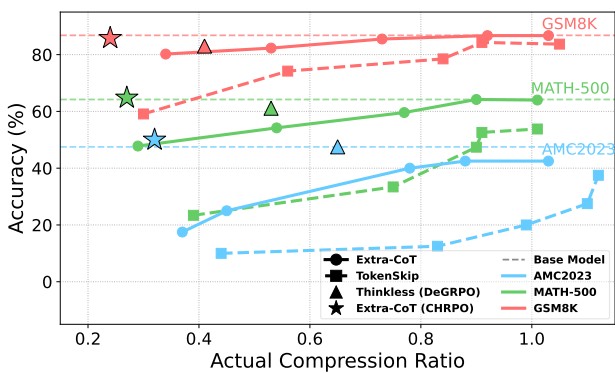

*Figure 1.* Comparison between accuracy and actual compression ratio of CoT tokens, defined as the ratio of the compressed CoT token length to the original length, across three math benchmarks evaluated on Qwen3-1.7B. Extra-CoT outperforms TokenSkip and Thinkless in the extremely low-ratio regime. CHRPO policy further improves performance at the lowest inference budgets, validating the effectiveness of our RL optimization.

## 1. Introduction

Large Language Reasoning Models (LRMs) (Yang et al., 2025; Huang et al., 2025; Comanici et al., 2025; Team et al., 2025), capable of generating step-by-step Chain-of-Thought (CoT) reasoning, have demonstrated remarkable performance in tasks requiring complex logical inference (e.g., OpenAI's o1 (Jaech et al., 2024) and DeepSeek-R1 (Guo et al., 2025)). By externalizing their reasoning, models can decompose intricate problems into manageable sub-steps, enhancing their problem-solving capabilities. However, this impressive performance requires a massive number of tokens. Models are prone to "overthinking", generating redundant reasoning paths even for simple queries (Chen et al., 2024; Sui et al., 2025; Su et al., 2025; Fan et al., 2025). This token-intensive generation leads to high computational overhead, hindering resource-limited applications.

Controllable CoT compression (e.g., TokenSkip (Xia et al., 2025) and CTS (Yuan et al., 2025)) has emerged by training models to follow specific budgets. These approaches, often relying on general importance estimators (e.g., LLMLingua-2 (Pan et al., 2024)), can successfully prune CoTs at low and

moderate levels (e.g., 50-60% of tokens). However, their performance often degrades catastrophically at high compression ratios (e.g., 20-30% tokens), as shown in Fig. 1. At these extreme levels, existing methods struggle to identify and preserve the sparse and critical reasoning steps, leading to a fatal loss of semantic integrity and logical fidelity. This loss of fidelity is critical, as recent studies (Wei et al., 2022; Jacovi et al., 2024) have demonstrated that the completeness and integrity of the reasoning chain are directly correlated with the final problem-solving ability. *It highlights a significant challenge that keeps semantically-preserved reasoning under the extreme-ratio CoT compression.*

To address the above challenge, we propose a novel Extreme-Ratio Chain-of-Thought Compression framework (*Extra-CoT*), which is designed to push the boundaries of CoT efficiency while maintaining high-accuracy reasoning via a tightly integrated pipeline, as shown in Fig. 2. The foundation of this pipeline is our novel semantically-preserved, question-aware compressor (§3.2), which is specifically designed to solve the *fidelity catastrophe*: The rapid performance degradation that general-purpose compressors exhibit at the extreme ratios (Yuan et al., 2025). It utilizes global attention to the input question and a novel index-based, formula-aware annotation method to generate supervision data, preserving the mathematical integrity. This semantically-preserved data is a prerequisite for the subsequent Mixed-Ratio SFT stage (§3.3). By training on the mixed-ratio, high-fidelity dataset, the SFT model learns to robustly adhere to ratio commands (e.g., $\langle \text{COMP\_40} \rangle$), instructing the model to be a specific target compression, thereby overcoming the poor adherence seen in the baselines. Simultaneously, this stage establishes the policy token ($\langle \text{COMP\_POLICY} \rangle$), which prompts the model to first release a ratio command as the trainable mechanism. Building upon this foundation of robust controllability, our novel RL algorithm, *CHRPO* (Constrained and Hierarchical Ratio Policy Optimization) (§3.4), is then applied to optimize this policy token. CHRPO acts as an extreme-compression optimizer with hierarchical rewards to explicitly incentivize the selection of the low budget while maintaining accuracy.

Extensive experiments on GSM8K (Cobbe et al., 2021), MATH-500 (Lightman et al., 2023) and AMC2023 (AI-MO Team, 2024) demonstrate the superiority of Extra-CoT. At the fixed ratios, our method significantly outperforms TokenSkip at higher compression levels. On MATH-500, Extra-CoT achieves 73% token reduction while improving accuracy by 0.6%. Moreover, at an extreme compression ratio of 0.2, our model maintains robust performance, whereas baseline methods suffer catastrophic accuracy collapse. Experimental results on GSM8K and AMC2023 present the same trend, underscoring the robustness of our approach in challenging mathematical reasoning benchmarks.

Our contributions can be summarized as:

1. We propose Extra-CoT, a novel three-stage framework for extreme-ratio CoT compression to achieve high-fidelity reasoning at the ultra-low budgets, which effectively alleviates the key challenges of semantic preservation and control adherence.

2. A novel semantically-preserved CoT compressor is proposed to utilize a question-aware architecture and a unique index-based, formula-aware annotation method, which generates high-fidelity supervision to preserve mathematical integrity.

3. A unified SFT-RL training pipeline is constructed, where a mixed-ratio SFT stage first provides robust, multi-ratio controllability and a trainable policy mechanism, followed by optimization with our hierarchical reward RL mechanism (CHRPO) for explicitly incentivizing accuracy at the extreme compression ratios.

**Conflict of Interest Disclosure.** W.L., J.H., and X.C. are affiliated with Huawei Foundation Model Dept.. This paper includes an auxiliary long-context evaluation using Pangu-Embedded-7B-V1.1. This relationship is disclosed for transparency in accordance with the ICML 2026 conflict-of-interest policy.

## 2. Related Work

**Prompt and Context Compression.** A primary path to efficiency is task-agnostic prompt compression. Methods like LLMLingua series (Jiang et al., 2023; 2024; Pan et al., 2024) and Selective Context (Li et al., 2023b) model this as an extractive task where a smaller model scores token importance. LLMLingua-2 (Pan et al., 2024), for example, uses data distillation to frame this as a token classification, achieving robust generalization. These methods provide a compressed prompt to a black-box LLM. Other works, such as Style-Compress (Pu et al., 2024) and CompAct (Yoon et al., 2024), explore generative prompt compression, generating abstractive compressed contexts rather than relying solely on extractive token selection. Our semantically-preserved compressor (§3.2) is built upon this extractive, token-classification paradigm.

**Extractive and Controllable CoT Compression.** A prominent path is pruning-based SFT, where an importance metric is used to prune CoTs. TokenSkip (Xia et al., 2025) observes varying token importance, prunes trajectories, and fine-tunes on a mixture of ratios to follow a given compression ratio $\gamma$. CTS (Yuan et al., 2025) refines this by using a reference model's perplexity to identify tokens critical to the final answer. Conditioned-prompt SFT (C3oT (Kang et al., 2025)) uses a powerful compressor (e.g., GPT-4) to

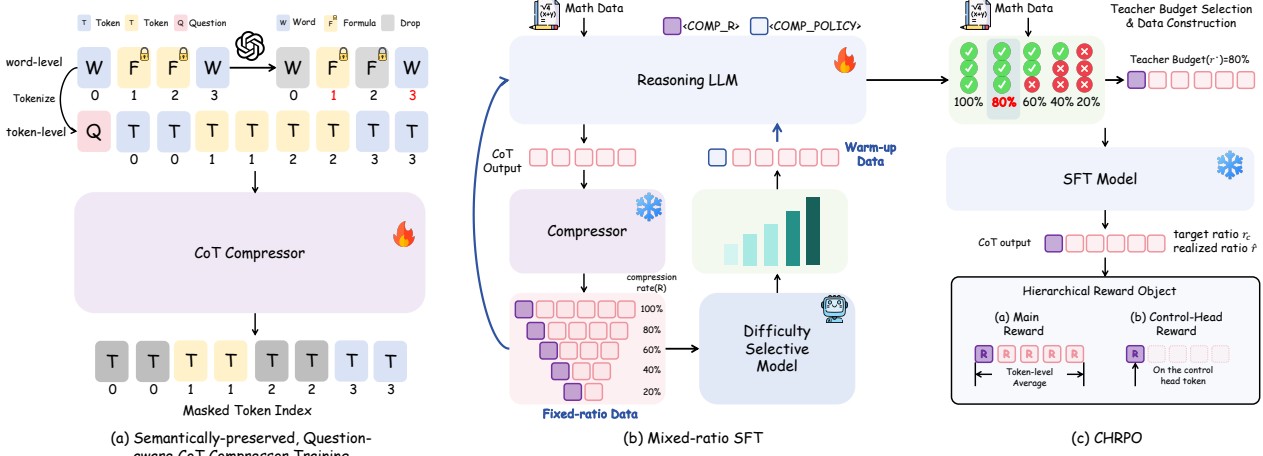

*Figure 2.* Overall pipeline of the proposed Extra-CoT, which includes three-stage training: (a) Semantically-preserved, question-aware CoT compressor training, (b) Mixed-ratio SFT and (c) CHRPO. We first train a CoT compressor on mathematical CoT data with fine-grained annotations to generate in-domain fixed-ratio compressed data. During mixed-ratio SFT stage, a reasoning LLM is fine-tuned on these fixed-ratio data combined with ratio-balanced warm-up data, teaching it to follow a spectrum of compression budgets and providing a stable initialization for the final stage. The final stage employs CHRPO to refine the model by using an accuracy-driven strategy to set teacher budgets and explicitly rewarding high accuracy in ultra-low compression regimes, thus incentivizing correct solutions.

create pairs of short and long summaries, and then fine-tunes the model with different prefix prompts, such that it can output either the short or the long version. This line of work, which fine-tunes models on compressed trajectories for controllability, is most relevant to our approach.

**Abstractive and Latent CoT Compression.** Beyond extractive pruning, some methods refactor the chain or move reasoning into latent space. Distill-Step-by-Step (Hsieh et al., 2023) treats LLM-generated rationales as supervision and trains a smaller, task-specific model in a multi-task manner. Sketch-of-Thought (Aytes et al., 2025) guides models to produce concise, structured sketches that avoid full-sentence elaboration. Latent CoT methods replace discrete intermediate text with continuous representations. CO-CONUT (Hao et al., 2024) introduces Chain-of-Continuous-Thought, which conducts intermediate reasoning in latent space, however, it relies on full-model fine-tuning, which leads to catastrophic forgetting. SoftCoT (Xu et al., 2025) mitigates this by freezing the backbone and using a small model to generate "soft thought" tokens, which are mapped into the backbone via a lightweight trainable projection.

**Budget-Aware and Adaptive Reasoning.** Parallel to deciding what to compress is deciding how much/when to think (Fang et al., 2025; Han et al., 2025; Wang et al., 2025; Hu et al., 2026). Thinkless decouples mode selection (short vs. long) from answer generation via DeGRPO; TALE predicts per-instance token budgets to steer decoding; and NOWAIT suppresses reflective tokens in inference to trim chains. These approaches chiefly regulate reasoning length/mode rather than directly constraining the extractive content and are complementary to our focus.

## 3. Methodology

### 3.1. Preliminaries

**Extractive CoT Compression.** Given a sample $(q, z, a)$, where $q$ is the question, $z = (t_1, \ldots, t_n)$ is the Chain-of-Thought (CoT) and $a$ is the answer, we frame CoT compression as an extractive selection problem. A compressor $C_\phi$ assigns an importance score $s_i$ to each token $t_i$:

$$s_i = \mathrm{Imp}_\phi(t_i \mid q, z) \in \mathbb{R}. \tag{1}$$

To meet a target ratio $\gamma \in (0, 1]$, we keep the top $k = \lfloor \gamma n \rfloor$ tokens, indexed by the set $\mathcal{K}_\gamma$. The compressed CoT is $\tilde{z}_\gamma = (t_i : i \in \mathcal{K}_\gamma)$. Two canonical instantiations for $\mathrm{Imp}_\phi$ are: (1) **Selective Context (PPL-based)** scores token surprisal using a causal LM:

$$s_i^{\mathrm{SC}} = -\log P_{\theta_{\mathrm{LM}}}(t_i \mid q, t_{<i}), \tag{2}$$

and (2) **Bi-encoder (Token-classification)** learns a keep-probability with a bidirectional encoder:

$$s_i^{\mathrm{BI}} = \Pr_{\theta_{\mathrm{BI}}}(y_i = 1 \mid q, z). \tag{3}$$

Previous works (Pan et al., 2024) show that the bi-encoder (Eq. (3)) often outperforms PPL-based methods (Eq. (2)) by alleviating position bias and capturing long-range dependencies (Pan et al., 2024). Thus, we adopt this extractive-on-CoT approach.

**Think-Only Accounting and Ratio Definitions.** To decouple the reasoning budget from answer formatting, we measure length strictly inside the reasoning segment. Let

$\tau(z)$ be the count of think-only tokens, and $L^* = \tau(z_{\text{full}})$ be the length of uncompressed CoT tokens $z_{\text{full}}$.

The real sample-level ratio is $\widehat{r} = \tau(\tilde{z}_\gamma)/L^* \in (0,1]$. We report answer accuracy and the dataset-level **ActRatio**, defined as the average sample-level ratio over dataset $\mathcal{D}$:

$$\text{ActRatio}(\gamma) \; = \; \mathbb{E}_{(q,z,a)\in\mathcal{D}}\left[\frac{\tau(\tilde{z}_\gamma)}{L^*}\right]. \qquad (4)$$

We use a matched-ratio protocol, comparing methods at the identical target ratios $\gamma$ and reporting their resulting $\text{ActRatio}(\gamma)$.

**Framework of Extra-CoT.** As demonstrated in Fig. 2, Extra-CoT has three stages: (1) a semantic compressor generates semantically-preserved data, (2) mixed-ratio SFT instills controllability and (3) CHRPO optimizes an autonomous policy. At inference, the model is governed by a control token $c_{\text{in}}$. If $c_{\text{in}}$ is $\langle\text{COMP\_POLICY}\rangle$, the CHRPO-optimized policy selects the ratio; if $c_{\text{in}}$ is a fixed-ratio token (e.g., $\langle\text{COMP\_40}\rangle$), the SFT-trained model adheres to that budget.

### 3.2. The Proposed CoT Compressor

**Semantically-Preserved Supervision Generation.** Our supervision pipeline begins by utilizing approximately 30K mathematical Chain-of-Thought (CoT) examples, generated by **Qwen3-32B** (Yang et al., 2025) on the CAMEL (Li et al., 2023a) dataset. For each training triple $(q, z, a)$, we first segment the CoT $z$ into word-level spans. We then detect all LATEX entities and inline mathematical expressions, collapsing each into a single, atomic unit. This step ensures formula atomicity, which is essential to prevent the compressor from fragmenting symbolic reasoning components.

Following this atomic segmentation, we generate robust supervision by prompting a teacher model (GPT-4o (Hurst et al., 2024)) with the question $q$ and the indexed CoT $z$, requiring it to return exclusively the set of indices to be preserved, $R \subseteq \{1, \ldots, m\}$ (detailed in Appendix A). This index-only approach is crucial for creating our semantically-preserved signal, as it inherently ensures robustness against minor paraphrasing or rendering variations in mathematical formulas. Finally, we align these span-level decisions back to tokens: tokens in kept spans receive label $y_i=1$, others receive $y_i=0$, producing exact, token-level supervision.

**Compressor Architecture and Objective Function.** We adopt Longformer-large-4096 (Beltagy et al., 2020) as the backbone. Crucially, we mark all tokens in the question $q$ with global attention, making the compressor question-aware. It allows every CoT token to directly attend to the question. The tokens within $z$ use local attention of the standard sliding-window. Let $x=[q;z]$ be the input. The final hidden state $\{h_i\}$ is passed to a linear classification head, which produces logits $o_i$ and probabilities $p_{i,c}$ for classes $c \in \{0{:}\text{DROP}, 1{:}\text{KEEP}\}$.

Let $\mathcal{I}_{\text{valid}}=\{i : y_i \in \{0,1\}\}$ be the set of valid CoT token indices. We train with a class-weighted Focal Loss, defined over the valid tokens as:

$$\mathcal{L}_{\text{Focal}} \; = \; -\mathbb{E}_{i\in\mathcal{I}_{\text{valid}}}\left[\alpha_{y_i}\left(1 - p_{i,y_i}\right)^\lambda \log p_{i,y_i}\right]. \quad (5)$$

Here, $\alpha_{y_i}$ is the class-weighting factor to balance positive/negative classes, and $\lambda \geq 0$ is the focusing parameter that reduces the loss contribution from high-confidence tokens, forcing the model to focus on hard examples.

**Compressor Decoding.** At inference, given a target ratio $\gamma$ and the token keep-scores $p_{i,1}$ from our compressor, we generate $\tilde{z}_\gamma$ using a greedy, length-aware selection. We sort tokens by $p_{i,1}$ and select them in descending order. This continues until the token budget $\lfloor \gamma L^* \rfloor$ is met, where $L^*$ is the original think-only length. The final $\tilde{z}_\gamma$ is the in-order concatenation of the kept spans/tokens.

### 3.3. Mixed-Ratio SFT

**Control Vocabulary and Prefix–Mirror Protocol.** To provide a unified interface for fixed-budget control and RL-based policy optimization, we augment the tokenizer with six special control tokens: $\langle\text{COMP\_20}\rangle$, $\langle\text{COMP\_40}\rangle$, $\langle\text{COMP\_60}\rangle$, $\langle\text{COMP\_80}\rangle$, $\langle\text{COMP\_100}\rangle$, and $\langle\text{COMP\_POLICY}\rangle$.

We introduce a Prefix–Mirror Protocol for the model's I/O format. Given an input $x = [q; c_{\text{in}}]$, where $q$ is the question and $c_{\text{in}}$ is a control token, the model is trained to generate an output $y = [c_{\text{out}}; \tilde{z}_\gamma]$, where $\tilde{z}_\gamma$ is the compressed rationale and the output prefix $c_{\text{out}}$ mirrors a ratio token based on one of the following two modes: 1) **Fixed-Ratio Mode.** If $c_{\text{in}}$ is a fixed-ratio token, the model is constrained to copy it verbatim, such that $c_{\text{out}} = c_{\text{in}}$; 2) **`<COMP_POLICY>` Warm-up Mode.** If $c_{\text{in}} = \langle\text{COMP\_POLICY}\rangle$, the model first autonomously predicts a ratio token $c_{\text{out}}$ and then generates the corresponding rationale $\tilde{z}_\gamma$.

**Data Construction.** Starting from **MetaMathQA-395K** (Yu et al., 2023), we use our CoT compressor to build an SFT dataset comprising two distinct cohorts according to the previous two modes: 1) **Fixed-Ratio Cohort** ($\mathcal{D}_{\text{fix}}$, 60k samples). We generate the compressed rationale $\tilde{z}_\gamma$ at each target ratio $\gamma \in \{0.2, \ldots, 1.0\}$ (12k samples per bucket) using our compressor; 2) **`<COMP_POLICY>` Warm-up Cohort** ($\mathcal{D}_{\text{policy}}$, 10k samples). We format data in the structure of $([q; \langle\text{COMP\_POLICY}\rangle], [c_{\text{out}}; \tilde{z}_\gamma])$, where the target bucket $c_{\text{out}}$ for each sample is determined by a heuristic-based difficulty-selective model (detailed in Appendix C).

**Pre-conditioning for Extreme Compression.** This SFT

stage is a critical pre-conditioning step that provides a stable foundation for the subsequent RL optimization. The two-cohort design serves distinct functions, both being essential to our Extra-CoT goal. In one hand, Fixed-Ratio Cohort ($\mathcal{D}_{\text{fix}}$) addresses the significant domain gap between general pre-training and our extreme compression target. By training the model on a spectrum of compression budgets—not just the target 0.2 ratio—we prevent performance collapse that would otherwise occur (detailed in Appendix C). This process renders the model with robust controllability and the ability to generate structurally-sound reasoning at low computation budgets. On the other hand, <COMP_POLICY> Cohort ($\mathcal{D}_{\text{policy}}$) serves as a complementary and critical function. While the fixed-ratio data teaches the model to follow deterministic compression commands, it provides no mechanism for the model to decide which ratio to choose. This cohort teaches the model to recognize <COMP_POLICY> as a trigger for its internal selection policy. This establishes the trainable mechanism for the RL stage, allowing CHRPO to be applied exclusively to the <COMP_POLICY> input with rewards steering the policy toward selecting extreme-low-budget outputs.

### 3.4. The Proposed CHRPO

**Setup & Notation.** Let $\mathcal{R} = \{0.2, 0.4, 0.6, 0.8, 1.0\}$ be the discrete set of target compression ratios. Given a query $x$, the policy $\pi_\theta$ first selects a target ratio $r_c \in \mathcal{R}$ via a control token and generates a reasoning trace enclosed within <think> tags followed by the final answer. The realized ratio, $\hat{r}$, is computed based exclusively on the token count of the reasoning trace to ensure strict budget counting.

We define two metrics to measure the ratio deviation. First, the selected ratio deviation, $g = k(r_c) - k(r^\star) \in \{-4, \ldots, 4\}$, quantifies the difference from the teacher budget $r^\star$, where $k(\cdot)$ maps a ratio to its grid index. A negative value for $g$ indicates the policy has chosen a more aggressive compression ratio than the teacher. Second, the deviation of the realized ratio, $\delta = \hat{r} - r_c$, measures the difference between the realized and selected target ratios.

**Teacher Budget Selection and Data Construction.** We construct a specialized dataset for RL from a disjoint subset of our data. To determine the teacher budget $r^\star$, for a given query $x$, we first run our SFT model across the entire discrete ratio grid, $r \in [0.2, \ldots, 1.0]$. We then identify the teacher budget $r^\star$ by applying a monotonically correct rule: $r^\star$ is defined as the smallest ratio $r$ for which the SFT model's answer is correct, provided that all larger ratios also yield correct answers. This monotonically correct check (detailed in the Appendix D) is critical for data quality. It ensures that we only select stable instances for RL and provides a reliable ground-truth budget for our reward shaping.

**Hierarchical Reward Objective.** The primary challenge

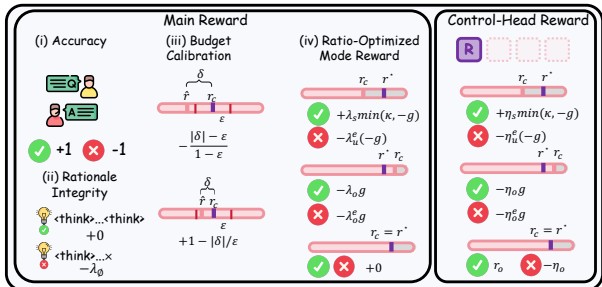

*Figure 3.* An illustration of our proposed CHRPO's hierarchical reward mechanism, which features a main reward and a control-head reward. The main reward, targeting all tokens, integrates four criteria: accuracy, rationale integrity, budget calibration, and rationale-optimized mode. In contrast, the control-head reward is applied only to the first token, providing a direct and immediate signal to shape the policy's ratio selection.

is to incentivize the policy to select extreme compression ratios while maintaining high accuracy. A single reward is ill-suited for this, as the crucial ratio selection at the first token is too distant from the final accuracy signal, leading to an unstable policy that avoids risk. To solve this, CHRPO implements a hierarchical reward structure (Fig. 3), applying two distinct rewards ($\mathcal{R}_{\text{main}}$, $\mathcal{R}_{\text{ctrl}}$) at different temporal locations, each targeting a specific sub-task.

**The Main Reward ($\mathcal{R}_{\text{main}}$).** The main reward, applied at the end of the sequence, targeting all tokens, consists of four components. The primary component is the Accuracy Reward, which evaluates the correctness of the final answer.

$$\mathcal{R}_{\text{acc}} = \mathbf{1}[\text{correct}] - \mathbf{1}[\neg\text{correct}] \in \{+1, -1\} \quad (6)$$

Second, we introduce the Ratio-Optimized Mode Reward ($\mathcal{R}_{\text{mode}}$), designed to explicitly incentivize the extreme low-budget regime while preventing the catastrophic drop in performance that can result from overly aggressive compression. Let $\kappa$ be a small cap (e.g., $\kappa = 2$). $\lambda_{\text{short}}$, $\lambda_{\text{over}}$, $\lambda_{\text{under}}^{\text{err}}$ and $\lambda_{\text{over}}^{\text{err}}$ act as coefficients controlling the magnitude of the reward or penalty given under four distinct scenarios.

$$\mathcal{R}_{\text{mode}}(g) = \begin{cases} +\lambda_{\text{short}} \, \min(\kappa, -g), & \text{if correct, } g < 0 \\ -\lambda_{\text{over}} \, g, & \text{if correct, } g > 0 \\ -\lambda_{\text{under}}^{\text{err}} \, (-g), & \text{if } \neg\text{correct, } g < 0 \\ -\lambda_{\text{over}}^{\text{err}} \, g, & \text{if } \neg\text{correct, } g > 0 \\ 0, & \text{otherwise.} \end{cases}$$

$$(7)$$

Third, we apply two constraints: Budget Calibration ($\mathcal{R}_{\text{cal}}$) using a Huber-like reward with tolerance $\varepsilon$ and Rationale Integrity ($\mathcal{R}_\varnothing$) with a penalty for failing to generate a

`<think>` block.

$$\mathcal{R}_{\text{cal}}(\delta) = \begin{cases} 1 - |\delta|/\varepsilon, & \text{if } |\delta| \leq \varepsilon \\ -\min\{1, \frac{|\delta|-\varepsilon}{1-\varepsilon}\}, & \text{if } |\delta| > \varepsilon \end{cases} \quad (8)$$

$$\mathcal{R}_{\varnothing} = -\lambda_{\varnothing} \mathbf{1}[\text{no-think}] \quad (9)$$

If no `<think>` block is present, $\mathcal{R}_{\text{cal}}$ is frozen to $0$ and only the $\mathcal{R}_{\varnothing}$ penalty applies, where the hyperparameter $\lambda_{\varnothing} > 0$ controls the magnitude of this penalty.

**The Control-Head Reward ($\mathcal{R}_{\text{ctrl}}$).** The control-head reward is applied exclusively at the first token to provide a direct and immediate gradient for the ratio selection policy. While $\mathcal{R}_{\text{main}}$ optimizes the subsequent execution of the rationale, $\mathcal{R}_{\text{ctrl}}$ focuses solely on the choice itself. It builds upon the risk-sensitive logic of $\mathcal{R}_{\text{mode}}$ but utilizes a separate set of coefficients and introduces a distinct reward $r_0$ for correctly matching the teacher budget:

$$\mathcal{R}_{\text{ctrl}}(g) = \begin{cases} r_0 + \eta_{\text{short}} \ \min(\kappa, -g), & \text{if correct, } g \leq 0 \\ -\eta_{\text{over}} \ g, & \text{if correct, } g > 0 \\ -\eta_{\text{under}}^{\text{err}} \ (-g), & \text{if } \neg\text{correct, } g < 0 \\ -\eta_0, & \text{if } \neg\text{correct, } g = 0 \\ -\eta_{\text{over}}^{\text{err}} \ g, & \text{if } \neg\text{correct, } g > 0. \end{cases} \quad (10)$$

**Final Objective and Rationale of Risk-Sensitive Shaping.**

The main reward aggregates weighted components for execution quality and constraints:

$$\mathcal{R}_{\text{main}} = \text{clip}_{[-1,1]}(w_{\text{acc}}\mathcal{R}_{\text{acc}} + w_{\text{cal}}\mathcal{R}_{\text{cal}} + \mathcal{R}_{\text{mode}} + \mathcal{R}_{\varnothing}) \quad (11)$$

The asymmetric penalties in Eq. (7) and (10) form the core of CHRPO's risk-sensitive strategy, providing three key guarantees stabilizing the policy's pursuit of extreme compression: 1) **Safe-Shortening Guarantee:** The policy is only rewarded for shortening ($g < 0$) when it is correct. If it shortens and fails, the large $\lambda_{\text{under}}^{\text{err}}$ penalty dominates the potential $\lambda_{\text{short}}$ gain. This ensures that the optimal policy only compresses truly compressible instances; 2) **Fail-Fast Recovery Guarantee:** The penalties are asymmetric ($\lambda_{\text{under}}^{\text{err}} > \lambda_{\text{over}}^{\text{err}}$). This means that the penalty for shortening-and-failing is larger than for being-too-long-and-failing. This gradient structure forces the policy to "fail safe": on hard instances, it prefers to up-shift to a longer budget rather than continuing to shorten, preventing a "stuck-at-short" policy collapse; 3) **Capped Progress:** The reward for successful compression is capped by $\kappa$. This encourages gradual and stable progress toward the extreme ratios, avoiding instability from large and discrete reward jumps.

## 4. Experiments

### 4.1. Experimental Setup

**Benchmarks.** We evaluate Extra-CoT on a comprehensive suite of mathematical reasoning benchmarks. Our primary results focus on GSM8K (Cobbe et al., 2021), MATH-500 (Lightman et al., 2023), and the AMC2023 benchmark (AI-MO Team, 2024). To ensure a thorough evaluation, we also test our method on SVAMP (Patel et al., 2021), MultiArith (Roy & Roth, 2015) and a 1k held-out test set from MetaMathQA-395K (Yu et al., 2023) (which we denote MetaMath-1k). Furthermore, to assess out-of-domain (OOD) generalization, our evaluation includes the STEM subset of MMLU (Hendrycks et al., 2020). The full results for SVAMP, MultiArith, MetaMath-1k, and all OOD benchmarks are provided in Appendix B.

**Backbone models.** Our primary results are demonstrated on Qwen3-1.7B (Yang et al., 2025) with a 4096-token context window. To examine the robustness of our compressed supervision across different backbones and context lengths, we further conduct evaluations on Qwen2.5-7B-Instruct (Yang et al., 2025), Llama3.2-3B-Instruct (Dubey et al., 2024), and Pangu-Embedded-7B-V1.1 (Chen et al., 2025). The Qwen2.5 and Llama experiments follow the short-context setting used in TokenSkip, while Pangu-Embedded-7B-V1.1 provides a 16K-context validation.

**Baselines.** We compare against four primary baselines: 1) Base Model, the uncompressed CoT generation from the backbone model; 2) Training-free methods, including length-control prompts and hard truncation of the output (Lee et al., 2025); 3) TokenSkip (Xia et al., 2025), our re-implementation using LLMLingua-2 as the compressor; 4) Thinkless* (Fang et al., 2025), our re-implementation trained using the proposed DeGRPO algorithm (details are provided in Appendix G).

**Implementation details.** All methods are implemented and evaluated under the same experimental pipeline for consistency. We fix a global random **seed (42)** for all data sampling, model initialization, and decoding to ensure reproducibility. To ensure a fair comparison, a single, consistent evaluation script is used for answer extraction and correctness verification across all methods and baselines. More training setups are presented in Appendix F.

**Evaluation protocol.** We evaluate all methods across five target compression ratios: $\gamma \in \{0.2, 0.4, 0.6, 0.8, 1.0\}$, and the $\langle \text{COMP\_POLICY} \rangle$ mode. Our evaluation centers on two primary metrics: accuracy and compression efficiency. Accuracy (Acc@all) is computed over the entire test set. To measure compression efficiency, we report the Actual Ratio (ActRatio), which is the realized compression ratio aggregated over the dataset. Crucially, all token counts and ratios are computed using think-only accounting, measur-

*Table 1.* Main comparison of SFT (TokenSkip, Extra-CoT) and RL (DeGRPO, Extra-CoT (CHRPO)) methods at matched target ratios on the Qwen3-1.7B backbone. Methods are evaluated across GSM8K, MATH-500 and AMC2023. The **best** and *second-best* scores are marked for each metric (lowest Tokens ↓, highest Acc@all ↑).

| Methods | Ratio | GSM8K | | | MATH-500 | | | AMC2023 | | |
|---|---|---|---|---|---|---|---|---|---|---|
| | | Tokens↓ | ActRatio | Acc@all↑ | Tokens↓ | ActRatio | Acc@all↑ | Tokens↓ | ActRatio | Acc@all↑ |
| **Base Model** | – | 873 | – | **86.8** | 1675 | – | *64.2* | 2092 | – | *47.5* |
| LC-Prompt (Lee et al., 2025) | 0.6 | 763 | 0.88 | 86.2 | 1466 | 0.88 | 60.4 | 2015 | 0.96 | 45.0 |
| Truncation (Lee et al., 2025) | 0.6 | 692 | 0.80 | 81.7 | 1271 | 0.76 | 49.8 | 1334 | 0.64 | 25.0 |
| TokenSkip (Xia et al., 2025) | 1.0 | 916 | 1.05 | 83.7 | 1696 | 1.01 | 54.4 | 2333 | 1.12 | 37.5 |
| TokenSkip (Xia et al., 2025) | 0.8 | 794 | 0.91 | 84.3 | 1527 | 0.91 | 53.0 | 2302 | 1.10 | 27.5 |
| TokenSkip (Xia et al., 2025) | 0.6 | 770 | 0.84 | 78.5 | 1517 | 0.90 | 47.4 | 2077 | 0.99 | 20.0 |
| TokenSkip (Xia et al., 2025) | 0.4 | 516 | 0.56 | 74.2 | 1259 | 0.75 | 34.4 | 1743 | 0.83 | 12.5 |
| TokenSkip (Xia et al., 2025) | 0.2 | *273* | 0.30 | 59.1 | 660 | 0.39 | 23.4 | 911 | 0.44 | 10.0 |
| TokenSkip (Xia et al., 2025) | <POLICY> | 473 | 0.52 | 70.0 | 1182 | 0.71 | 35.8 | 1437 | 0.69 | 12.5 |
| **Extra-CoT** | 1.0 | 902 | 1.03 | *86.7* | 1698 | 1.01 | 64.0 | 2153 | 1.03 | 42.5 |
| **Extra-CoT** | 0.8 | 807 | 0.92 | *86.7* | 1520 | 0.90 | *64.2* | 1845 | 0.88 | 42.5 |
| **Extra-CoT** | 0.6 | 641 | 0.73 | 85.5 | 1304 | 0.77 | 59.6 | 1624 | 0.78 | 40.0 |
| **Extra-CoT** | 0.4 | 469 | 0.53 | 82.3 | 920 | 0.54 | 54.2 | 945 | 0.45 | 25.0 |
| **Extra-CoT** | 0.2 | 303 | 0.34 | 80.2 | *481* | 0.29 | 47.8 | *782* | 0.37 | 17.5 |
| **Extra-CoT** | <POLICY> | 569 | 0.65 | 86.3 | 1312 | 0.78 | 63.0 | 1515 | 0.72 | 42.5 |
| **Thinkless* (DeGRPO)** (Fang et al., 2025) | – | 356 | 0.41 | 85.5 | 888 | 0.53 | 63.6 | 1369 | 0.65 | **50.0** |
| **Extra-CoT (CHRPO)** | <POLICY> | **210** | 0.24 | 85.8 | **452** | 0.27 | **64.8** | **675** | 0.32 | **50.0** |

ing tokens strictly within <think>...</think> blocks. Consequently, ActRatio and its underlying token counts are averaged over parsable outputs.

## 4.2. Main Results

Table 1 presents the main results between Extra-CoT and other baselines. Extra-CoT consistently and significantly outperforms TokenSkip across all benchmarks and ratios.

**Performance at the matched ratios.** The performance gap is most pronounced in aggressive compression budgets ($\gamma \leq 0.4$), where symbolic fidelity and question-alignment are critical. On MATH-500 at $\gamma = 0.2$, our method achieves 47.8 Acc@all, a +24.4 point gain over TokenSkip. This substantial lead is maintained at $\gamma = 0.4$. On GSM8K, Extra-CoT achieves an accuracy of 80.2 at $\gamma = 0.2$, surpassing TokenSkip by +21.1 points. Similarly, on AMC2023, we lead by +7.5 points at $\gamma = 0.2$. Even at a modest $\gamma = 0.8$, our method provides a +2.4 point gain on GSM8K, demonstrating superior reasoning ability on all compression ratios.

**Comparison of optimized policies.** We compared our Extra-CoT (CHRPO) policy with the RL-based Thinkless (DeGRPO) baseline. On MATH-500, our policy (Table 1) achieves 64.8% accuracy at a highly compressed 0.27 ActRatio, outperforming Thinkless by +1.2 points. On GSM8K, it matches Thinkless's accuracy while using fewer tokens. A similar pattern holds on AMC2023, where our policy achieves 50.0% accuracy at the 0.32 ActRatio, matching Thinkless but with less than half the tokens, demonstrating a clearly superior accuracy-efficiency frontier. This practical efficiency gain is further reflected in the end-to-end latency results in Table 5.

**Ratio adherence and control collapse.** Beyond accuracy,

Table 1 reveals a critical difference in ratio adherence. Both SFT models exhibit a positive deviation (where ActRatio > TargetRatio), as the model must trade off between command-following and logical correctness. However, this deviation is far more severe for TokenSkip, suffering an evident "control collapse" on MATH-500: its $\gamma = 0.8$ and $\gamma = 0.6$ targets produce nearly identical realized ratios. This trend holds on GSM8K, where $\gamma = 0.6$ target overshoots to ∼0.84. In contrast, Extra-CoT maintains clear and predictable control, achieving a much closer ActRatio of ∼0.73 on GSM8K and clearly distinguishing the 0.8 and 0.6 targets on MATH-500 (0.90 vs. 0.77). This quantitative difference suggests that the SFT model learns that the baseline's low-ratio supervision data is "untrustworthy", causing it to disobey the command and default to a safer, longer output. Conversely, Extra-CoT's high-fidelity data empowers the model to follow extreme-ratio commands with greater fidelity, as our qualitative analysis will demonstrate.

## 4.3. Analysis and Ablations

**Qualitative Analysis at Extreme Ratios.** To qualitatively assess the impact of our semantically-preserved compressor, we present a side-by-side comparison of its outputs against those from the LLMLingua-2 on GSM8K (Fig. 4). The comparison, shown at 0.2 and 0.4 compression ratios, highlights different preservation strategies. The baseline, a general-purpose compressor, successfully shortens the text but shows a tendency to prune tokens within symbolic expressions, leading to partial fragmentation of formulas. In contrast, our specialized compressor demonstrates a superior preservation of mathematical integrity. It consistently identifies and retains entire formulas as atomic units, a direct result of our formula-aware annotation. This case study highlights that our compressor is beneficial for maintaining

| Our Compressor | LLMLingua-2 | Our Compressor | LLMLingua-2 |
|---|---|---|---|
| ...Starting equation $$x^2 + 12x = 73$$ To complete the square, I need to ... The coefficient of $x$ is 12, so half of that is 6, and squaring it gives 36. $$x^2 + 12x + 36 = 73 + 36$$ $$x^2 + 12x + 36 = 109$$ $$(x +6)^2 = 109$$ Taking the square root...:\n$$x + 6 = \\pm\\sqrt{109}$$ = -6 pm $$x = -6 + \\sqrt{109}$$ . -6 + sqrt{109} which is sqrt{109}- 6 a b a + b = 109 + 6 = 115 ... Therefore, $a = 109$ and $b = 6$, so $a + b = 115$. 2 (- ) ... Substituting $b = 6$ into the second equation: $a + 6^2 - 12*6 = 73$\n$a + 36- 72 = 73$\n$a - 36 = 73$\n$a = 73 + 36 = 109$ Therefore, $a + b = 109 + 6 =115$. | equation^2 + 12x = 73 positive solution{a} - b + b quadratic equations^2 + 12x = 73 number sides square trinomial coefficient 12 6 36. 73 109 square +^2 = 109+ 6 =\\sqrt{109} -6 = -6 + \\sqrt{109}\\sqrt{a} - b-6 + 6 115 109 6 ...factors 1 109 + 6 = 115{a - b^2 + 12x - 73 73 prime unlikely 144 + 292 = 436 perfect factor form - - b +12b 73 = 73 coefficients equal coefficient zero constant 73 + - 12b 73 equation 6^2 - 12*6 = 73 - 109 $b 6 + b 109 + 6 = 115 answer 115 $x{109 - 6 equation Compute^2 +{109 - 6 145 - 126) = 72 73 73original equation solution correct = 109 = 6 = 115 mistakes answer 115 | we need to find all real numbers $ k $ for which there's a nonzero 2-dimensional vector $ \\mathbf{v} $ such that the matrix multiplied by $\\mathbf{v} $ equals $ k \\mathbf{v} $. First, I remember that..., that means... So, given that the matrix... eigenvalues of a matrix are found by...$ \\det(A - kI) = 0 $...Calculating the determinant...Which simplifies to...Then, the characteristic equation...Therefore, $ k = 1 \\pm 4 $, so k = 5 or k = -3. So, yes, -3 is indeed an eigenvalue...But the characteristic equation is (1 - k)^2 - 8X = 0. So, if k = -3, then (1 - (-3))^2 - 8X = 0 => 16 - 8X = 0 => X = 2. | ...\\begin 1 & 8 \\\\\\\ X & 1...multiplied equals k answer -3...eigenvalue k problem eigenvalues matrix equation \"find real numbers $ k nonzero vector $ \\mathbf{v} eigenvalues problem answer -3...original problem find k -3 asking X -3 eigenvalue answer X = 2. value X 2...has eigenvalue k eigenvalues solutions characteristic equation -3 eigenvalue find X characteristic equation...another one problem answer original question find k answer -3 user asking X eigenvalue answer X = 2. X not 2 equation (1 - k)^2 = 8X different solutions -3 eigenvalue substituting k - 3 gives X 2. X 2... |

(a) R=0.2      (b) R=0.4

*Figure 4.* Comparison of output quality between our compressor and LLMLingua-2 at 0.2 and 0.4 compression ratios. While our compressor produces a coherent and semantically faithful output that preserves structural and formula integrity, LLMLingua-2's output degrades into a fragmented text with semantic discontinuities and incomplete formulas.

*Table 2.* SFT-level comparison of Extra-CoT (Ours) and TokenSkip on the GSM8K benchmark, evaluated on the Qwen2.5-7B-Instruct and Llama-3.2-3B-Instruct backbones.

| Methods | Ratio | Qwen2.5-7B-Instruct | | | Llama3.2-3B-Instruct | | |
|---|---|---|---|---|---|---|---|
| | | Tokens↓ | ActRatio | Acc↑ | Tokens↓ | ActRatio | Acc↑ |
| **Base Model** | – | 297 | – | 91.4 | 214 | – | 79.5 |
| **TokenSkip** | | | | | | | |
| 1.0 | | 296 | 1.00 | 91.7 | 199 | 1.00 | *80.5* |
| 0.9 | | 255 | 0.86 | 91.1 | 182 | 0.90 | 78.0 |
| 0.8 | | 237 | 0.80 | 90.1 | 162 | 0.80 | 77.0 |
| 0.7 | | 217 | 0.73 | 89.9 | 153 | 0.70 | 74.1 |
| 0.6 | | 178 | 0.60 | 87.9 | 141 | 0.60 | 73.9 |
| 0.5 | | **151** | 0.51 | 86.0 | *123* | 0.50 | 72.5 |
| **Extra-CoT (Ours)** | | | | | | | |
| 1.0 | | 298 | 1.00 | **92.3** | 196 | 1.00 | **81.5** |
| 0.9 | | 283 | 0.95 | *92.1* | 181 | 0.90 | 79.3 |
| 0.8 | | 261 | 0.87 | 91.7 | 163 | 0.80 | 79.2 |
| 0.7 | | 236 | 0.79 | 90.5 | 145 | 0.70 | 78.5 |
| 0.6 | | 222 | 0.71 | 89.9 | 132 | 0.60 | 77.0 |
| 0.5 | | *189* | 0.63 | 89.4 | **115** | 0.50 | 73.4 |

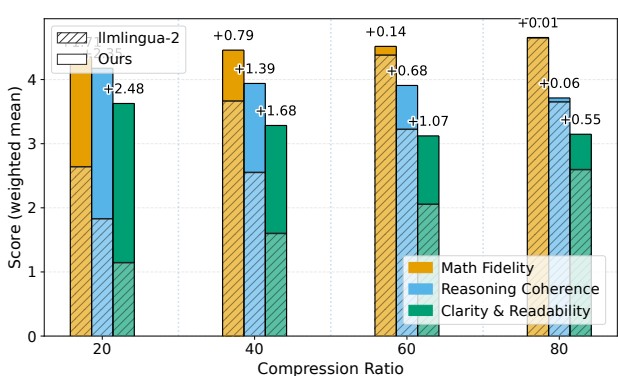

*Figure 5.* Compressor quality comparison between our method (Ours) and LLMLingua-2. Both compressors were used to compress the same dataset at four fixed compression ratios. LLMs then scored the outputs on a 1-5 scale across three metrics: **Math Fidelity**, **Reasoning Coherence**, and **Clarity & Readability**.

logical coherence under high compression.

**Quantitative Compressor Evaluation.** To quantify compressor quality and test our hypothesis that low-fidelity supervision causes the control collapse, we used Deepseek-R1 (Guo et al., 2025) and Qwen3-32B (Yang et al., 2025) as objective judges to score our compressor against LLMLingua-2 (Pan et al., 2024) (detailed in Appendix A). As shown in Fig. 5, outputs at four ratios were scored 1–5 on Math Fidelity, Reasoning Coherence, and Clarity & Readability. Our compressor outperforms LLMLingua-2, with the largest gaps in fidelity and coherence observed at the extreme 0.2 and 0.4 ratios. To verify that this trend is not merely an artifact of LLM-based judging, we further conduct a blind A/B human preference study. Five expert raters compare compressed rationales from Extra-CoT and LLMLingua-2 across four target budgets, yielding 250 pairwise votes per budget and 1,000 votes in total. The human results show the same trend as the automatic evaluation, with Extra-CoT being strongly preferred under aggressive

compression (detailed in Appendix H).

**Effect of RL Reward Components.** The ablation study (Table 3) validates the necessity of both the mode shaping ($\mathcal{R}_{\text{mode}}$) and the control-head ($\mathcal{R}_{\text{ctrl}}$) components in CHRPO. Removing both (w/o $\mathcal{R}_{\text{mode+ctrl}}$) causes the GSM8K average token count to surge from the target 210 to 324, confirming that $\mathcal{R}_{\text{mode}}$ is the primary engine driving the policy toward the extreme low-budget regime. Conversely, ablation of the control-head (w/o $\mathcal{R}_{\text{ctrl}}$) results in significant accuracy degradation. This confirms that $\mathcal{R}_{\text{ctrl}}$'s hierarchical placement is essential for stabilizing the policy and ensuring the aggressive compression remains high-quality. Only the full CHRPO framework achieves the optimal balance of maximum compression efficiency (210 Tokens) and highest accuracy (85.8%). Further ablation studies are provided in Appendix E.

**Robustness Across Backbones and Context Lengths.** To

*Table 3.* Ablation study on the core CHRPO reward components. Performance of the **Full CHRPO** model is compared against two ablated versions: lacking only the control-head reward (w/o $\mathcal{R}_{\text{ctrl}}$) and lacking both mode and control rewards (w/o $\mathcal{R}_{\text{mode+ctrl}}$).

| Ablation | GSM8K | | MATH-500 | | MetaMath-1K | |
|---|---|---|---|---|---|---|
| | Tkn ↓ | Acc ↑ | Tkn ↓ | Acc ↑ | Tkn ↓ | Acc ↑ |
| w/o $\mathcal{R}_{\text{mode+ctrl}}$ | 324 | *85.7* | 604 | *60.0* | 334 | **91.2** |
| w/o $\mathcal{R}_{\text{ctrl}}$ | *258* | 82.1 | *568* | 59.6 | *267* | 89.0 |
| **Full CHRPO** | **210** | **85.8** | **452** | **64.8** | **213** | *91.1* |

*Table 4.* Long-context validation on Pangu-Embedded-7B-V1.1 with a 16K context window. We report average generated tokens and Acc@all on GSM8K and MATH-500.

| Method | Ratio | GSM8K | | MATH-500 | |
|---|---|---|---|---|---|
| | | Tok. ↓ | Acc. ↑ | Tok. ↓ | Acc. ↑ |
| Base Model (Auto) | – | 2271 | 83.2 | 5511 | 77.2 |
| Base Model (Fast) | – | 1554 | 75.7 | 4954 | 72.2 |
| Extra-CoT | 0.2 | 1021 | 80.7 | 2822 | 66.0 |
| Extra-CoT | 0.4 | 1720 | 83.7 | 3755 | 74.6 |
| Extra-CoT | 0.6 | 2040 | 82.6 | 4259 | 77.4 |
| Extra-CoT | 0.8 | 2192 | 84.5 | 4748 | 79.4 |
| Extra-CoT | `<POLICY>` | 2070 | 84.2 | 3827 | 74.0 |

validate whether the benefits of our compressed supervision are consistent across different backbone models and context regimes, we conduct additional evaluations beyond the primary Qwen3-1.7B setting. We first replicate the short-context ($< 1024$) setting of TokenSkip using Qwen2.5-7B-Instruct and Llama3.2-3B-Instruct. As shown in Table 2, replacing the baseline compressor with our question-aware and formula-preserving compressor yields consistent gains at matched compression ratios, especially under lower budgets. This indicates that the quality of compressed supervision is a key factor behind the observed improvements, rather than an artifact of a specific backbone.

We further evaluate Extra-CoT on Pangu-Embedded-7B-V1.1 with a 16K context window. As shown in Table 4, Extra-CoT maintains a favorable accuracy–efficiency trade-off on both GSM8K and MATH-500 under a substantially larger context limit. On GSM8K, the `<POLICY>` mode achieves 84.2 Acc@all with 2070 generated tokens, outperforming the Auto preset of the base model while using fewer tokens. At the most aggressive ratio, Extra-CoT also surpasses the Fast preset by 5.0 accuracy points while generating fewer tokens. On MATH-500, Extra-CoT at ratio 0.8 improves over the Auto preset by 2.2 accuracy points with fewer tokens, while the 0.4 and `<POLICY>` modes both provide better accuracy–efficiency trade-offs than the Fast preset. These results suggest that Extra-CoT does not merely exploit short-context constraints, but remains effective on long-context reasoning backbones.

*Table 5.* End-to-end inference latency (seconds per instance, lower is better) on three reasoning benchmarks. All methods are evaluated under the same decoding configuration. The **best** and *second-best* scores are marked for each dataset (lowest Latency ↓).

| Dataset | Base Model | TokenSkip | Extra-CoT (Ours) |
|---|---|---|---|
| GSM8K | 0.7298 | *0.6829* | **0.2254** |
| MATH-500 | *1.8176* | 1.9409 | **0.7721** |
| MetaMath-1k | 0.7186 | *0.6769* | **0.2799** |

**End-to-End Latency.** Since the goal of CoT compression is practical reasoning efficiency, we further measure end-to-end per-instance inference latency under the same decoding configuration. For a favorable comparison, TokenSkip is evaluated at its smallest compression ratio, while Extra-CoT uses the CHRPO-trained `<POLICY>` mode. As shown in Table 5, Extra-CoT consistently translates token savings into real wall-clock speedups, reducing latency by $3.24\times$ on GSM8K, $2.35\times$ on MATH-500, and $2.57\times$ on MetaMath-1k compared with the base model. In contrast, TokenSkip provides only marginal latency gains on GSM8K and MetaMath-1k and is slower than the base model on MATH-500, suggesting that low-fidelity compression does not reliably translate into runtime improvements.

## 5. Conclusion

We present Extra-CoT, a three-stage framework that improves the efficiency of CoT reasoning while preserving fidelity under extreme compression. It combines (i) a formula-aware compressor for supervision, (ii) mixed-ratio SFT to enable robust controllability, and (iii) CHRPO, a hierarchical RL algorithm for ultra-low token budgets. Experiments show that Extra-CoT consistently outperforms TokenSkip at high compression ratios, indicating that high-fidelity extreme compression is a viable path to efficient reasoning.

## Acknowledgements

This work is supported by the National Natural Science Foundation of China (NO. 62572193), China Postdoctoral Science Foundation (NO. 2024M760930), the Open Research Fund of the Key Laboratory of Advanced Theory and Application in Statistics and Data Science, Ministry of Education, and the Fundamental Research Funds for the Central Universities.

## Impact Statement

This paper presents work whose goal is to advance the field of machine learning. There are many potential societal consequences of our work, none of which we feel must be specifically highlighted here.

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

*Figure 6.* Compression Labeling Prompt used to generate supervision data.

*Figure 7.* Compression Evaluation Prompt.

## A. Prompt Templates for Compression and Evaluation

**Compression Labeling Prompt.** We employ a specialized prompt to leverage GPT-4o as our primary annotator for CoT compression. Provided with a question and a word-indexed CoT, the model is tasked with identifying the minimal subsequence of token indices necessary to reconstruct a complete, question-specific reasoning path. The prompt enforces two structural constraints: (1) `[MATH_k]` placeholders must be preserved as atomic units to maintain symbolic integrity, and (2) conversational filler, reasoning dead ends, and redundant steps must be eliminated. To facilitate programmatic parsing, the output is strictly constrained to a JSON format containing a list of ascending, non-overlapping index intervals. The full prompt template is illustrated in Fig. 6.

**Compression Evaluation Prompt.** To quantify compressor quality, we utilize DeepSeek-R1 and Qwen3-32B as an objective judge to evaluate compressed CoT candidates. The model operates in a pairwise comparison setting or a direct scoring setting, assigning scores (1–5) across three specific dimensions: **Math Fidelity**, **Reasoning Coherence**, and

**Clarity & Readability**. The judge is explicitly instructed to verify whether the compressed text retains the logical validity of the original solution. The output is structured as a JSON object containing individual scores and a brief justification to facilitate automated aggregation and statistical analysis. The complete evaluation prompt is presented in Fig. 7.

## B. Additional Experiments

**Performance on In-Domain Benchmarks.** As reported in Table 6, Extra-CoT maintains a significant performance advantage on in-domain benchmarks, with the gap becoming most pronounced at aggressive compression ratios. On **SVAMP**, our method achieves 87.3 Acc@all at $\gamma = 0.2$, decisively outperforming TokenSkip by **+21.3** points. This substantial lead is mirrored on **MetaMath-1k**, where Extra-CoT (86.0%) surpasses TokenSkip (68.4%) by **+17.6** points at the same ratio. Even on the simpler **MultiArith** benchmark, where the uncompressed Base Model approaches saturation, TokenSkip's accuracy notably degrades at $\gamma = 0.2$ (91.1%), whereas our method maintains parity with the baseline at 97.8%, demonstrating superior fidelity in preserving critical reasoning steps.

**Robustness on Out-of-Domain (OOD) Tasks.** We observe a similar trend on the **MMLU (STEM)** benchmark, which tests generalization to unseen and more complex problems. While extreme compression on this OOD task expectedly impacts absolute accuracy (dropping from 74.1% Base Model to 54.3% at $\gamma = 0.2$), Extra-CoT proves significantly more robust than the baseline. At $\gamma = 0.2$, TokenSkip suffers a catastrophic collapse to 18.1 Acc@all. In stark contrast, our method maintains 54.3% accuracy, securing a massive **+36.2** point lead. This underscores that while OOD generalization at extreme ratios remains a challenging frontier, our semantically-aware framework provides a critical safety buffer against the complete reasoning failure observed in existing methods.

*Table 6.* **Full evaluation on additional benchmarks.** Performance is reported on **Qwen3-1.7B** across in-domain tasks (SVAMP, MultiArith, MetaMath-1k) and the Out-of-Domain (OOD) benchmark MMLU (STEM). Methods include SFT (TokenSkip, Extra-CoT) and RL (Extra-CoT (CHRPO)). The lowest Tokens ↓ and highest Acc@all ↑ are desirable.

| Methods | Ratio | SVAMP | | | MultiArith | | | MetaMath-1k | | | MMLU (STEM) (OOD) | | |
|---|---|---|---|---|---|---|---|---|---|---|---|---|---|
| | | Tokens↓ | ActRatio | Acc↑ | Tokens↓ | ActRatio | Acc↑ | Tokens↓ | ActRatio | Acc↑ | Tokens↓ | ActRatio | Acc↑ |
| **Base Model** | – | 608 | – | **93.3** | 320 | – | 98.3 | 823 | – | 91.0 | 1081 | – | **74.1** |
| TokenSkip | 1.0 | 697 | 1.15 | 90.0 | 323 | 1.00 | **99.4** | 878 | 1.07 | 91.1 | 1556 | 1.44 | 56.0 |
| TokenSkip | 0.8 | 541 | 0.89 | 90.3 | 266 | 0.83 | 98.9 | 751 | 0.85 | 90.0 | 1481 | 1.37 | 54.5 |
| TokenSkip | 0.6 | 502 | 0.83 | 85.7 | 260 | 0.81 | 96.1 | 765 | 0.87 | 87.9 | 1310 | 1.21 | 47.5 |
| TokenSkip | 0.4 | 338 | 0.56 | 80.3 | 177 | 0.55 | 94.4 | 531 | 0.60 | 83.3 | 877 | 0.81 | 29.8 |
| TokenSkip | 0.2 | 214 | 0.35 | 66.0 | 108 | 0.34 | 91.1 | 313 | 0.37 | 68.4 | 446 | 0.41 | 18.1 |
| TokenSkip | <POLICY> | 357 | 0.59 | 75.3 | 175 | 0.55 | 95.0 | 525 | 0.60 | 78.4 | 1035 | 0.96 | 32.6 |
| **Extra-CoT** | 1.0 | 656 | 1.07 | 93.0 | 328 | 1.00 | 98.3 | 876 | 1.06 | **92.4** | 1252 | 1.16 | 70.5 |
| **Extra-CoT** | 0.8 | 612 | 1.00 | 93.0 | 278 | 0.87 | 98.9 | 796 | 0.96 | **92.4** | 1267 | 1.17 | 71.3 |
| **Extra-CoT** | 0.6 | 473 | 0.78 | 93.0 | 214 | 0.67 | 98.3 | 634 | 0.77 | 90.6 | 1012 | 0.94 | 60.6 |
| **Extra-CoT** | 0.4 | 301 | 0.49 | 90.0 | 173 | 0.54 | 96.7 | 458 | 0.53 | 89.0 | 730 | 0.67 | 61.3 |
| **Extra-CoT** | 0.2 | 208 | 0.34 | 87.3 | 116 | 0.36 | 97.8 | 292 | 0.33 | 86.0 | 458 | 0.42 | 54.3 |
| **Extra-CoT** | <POLICY> | 341 | 0.52 | 90.0 | 168 | 0.53 | 97.8 | 569 | 0.69 | 91.6 | 1026 | 0.95 | 67.5 |
| **Extra-CoT (CHRPO)** | <POLICY> | **127** | 0.21 | 91.8 | **90** | 0.28 | **99.4** | **212** | 0.25 | 91.1 | **330** | 0.31 | 67.1 |

# C. Policy-Mode SFT: Budget Design and Difficulty Selection

**Why Five Fixed Ratios** We hypothesize that a denser ratio curriculum is essential for stable learning at extreme budgets. To test this, we compare our default five-ratio mixture $\{0.2, 0.4, 0.6, 0.8, 1.0\}$ against a sparse, endpoints-only mixture $\{0.2, 1.0\}$ under an identical SFT setup. The five-ratio model achieves an accuracy of 53.9% on MetaMath-1k when evaluated at the extreme $r=0.2$ ratio, whereas the endpoints-only model yields only 47.1%. We posit that the intermediate budgets ($\{0.4, 0.6, 0.8\}$) act as critical **semantic anchors**. These anchors regularize the model's internal mapping from a target ratio command $r$ to its generation policy, providing a smoother curriculum between full copying ($r=1.0$) and extreme compression ($r=0.2$). This dense supervision reduces label discontinuities and the variance of token-level decisions, thereby improving the model's calibration and fidelity precisely at the most aggressive, hardest-to-learn budget ($r=0.2$).

**Difficulty-Selective Heuristic** To construct the $\mathcal{D}_{\text{policy}}$ cohort for SFT, we developed a lightweight heuristic to stratify problems into five difficulty tiers using only their gold CoT. This approach avoids any costly external model calls. For every CoT, we compute four simple signals that correlate strongly with the cognitive reasoning burden: (i) Total **Length** ($L$); (ii) **Equation/Step Count** ($E$), serving as a proxy for procedural complexity; (iii) **Operator Richness** ($O$), measuring symbolic diversity; and (iv) **Lexical Richness** ($R$), which counts the unique reasoning concepts to capture the semantic diversity of the problem.

These signals are normalized per-dataset (yielding $\tilde{L}, \tilde{E}, \tilde{O}, \tilde{R}$) and aggregated into a weighted score $S$:

$$S = 0.35\tilde{L} + 0.25\tilde{E} + 0.20\tilde{O} + 0.20\tilde{R} \quad (12)$$

Tiers are then assigned via 20/40/60/80% quantile cuts on $S$, producing balanced five-level difficulty labels. These labels are subsequently used to construct the $\langle\text{COMP\_POLICY}\rangle$ warm-up data, specifically to ensure the policy is initialized with a balanced distribution of problem difficulties across varying compression targets.

# D. Reinforcement Learning Data Pipeline

We construct the RL training dataset $\mathcal{S}$ by post-processing the inference trajectories of the SFT model. For every problem instance indexed by id, we evaluate the model accuracy $P_{\text{map}}[r]$ across the entire discrete ratio grid $\mathcal{R} = \{0.2, \ldots, 1.0\}$. To ensure high-quality supervision, a "teacher budget" $r^\star$ is assigned to each id based on a strict **Monotonic Correctness Criterion**. Specifically, $r^\star$ is defined as the smallest (most compressed) ratio $r$ such that the model answers correctly at $r$ *and* at all looser constraints $r' > r$. Any instance id that violates this monotonicity (e.g., correctly solved at $r=0.2$ but failed at $r=0.6$) is deemed unstable and discarded. This filtering step is critical for eliminating lucky guesses and ensuring a reliable reward signal.

The retained, monotonically-correct examples are then grouped into buckets $C_r$ according to their assigned $r^\star$. To construct the final dataset $\mathcal{S}$, we sample from each bucket to satisfy a target quota $Q_r$, while enforcing global deduplication to ensure each unique problem id appears at most once in the training set. The complete sampling procedure is summarized in Algorithm 1.

# E. Reward Design and Ablations

We extend the analysis from the main text by evaluating the isolation effects of the **Fixed-Ratio Cohort** during SFT and the **Calibration Reward** ($\mathcal{R}_{\text{cal}}$) during RL. The results are

**Algorithm 1** RL Data Sampling

**Require:**

    Data Dirs $D_r$ and Quotas $Q_r$

1: RATIOS $\leftarrow [0.2, 0.4, 0.6, 0.8, 1.0]$
2: Load all files $D_r$; keep one entry per $(id, r)$
3: $D_{1.0} \leftarrow$ entries at $r=1.0$
4: Init buckets $C_r$ for $r \in$ RATIOS
5: **for all** $id$ in $D_{1.0}$ **do**
6:     $L_{ref} \leftarrow$ tokens(`<think>` at $r=1.0$ for $id$)
7:     **if** $L_{ref} \leq 0$ **then continue**
8:     **end if**
9:     $P_{map}, L_{map}$                          $\leftarrow$
    BUILDPASSANDMAPS($id$, RATIOS)
10:     $r^\star \leftarrow$ FINDMONOTONICTARGET($P_{map}$, RATIOS)
11:     **if** $r^\star$ is None **then continue**
12:     **end if**
13:     $row \leftarrow$ CREATEROWDATA($id, r^\star, L_{ref}, L_{map}[r^\star]$)
14:     $C[r^\star]$.append($row$)
15: **end for**
16: $\mathcal{S} \leftarrow \emptyset, \mathcal{U} \leftarrow \emptyset$          ▷ Final Set, Used IDs
17: S_count $\leftarrow$ new Map() with default 0
18: **for** $r \in$ RATIOS **do**
19:     SHUFFLE($C[r]$)
20:     **for all** $row \in C[r]$ **do**
21:         **if** S_count$[r] < Q[r]$ **and** $row.id \notin \mathcal{U}$ **then**
22:             $\mathcal{S}$.add($row$)
23:             $\mathcal{U}$.add($row.id$)
24:             S_count$[r] \leftarrow$ S_count$[r] + 1$
25:         **end if**
26:     **end for**
27: **end for**

summarized in Table 7.

**Impact of SFT Fixed-Ratio Cohort.** Removing the Fixed-Ratio Cohort ($\mathcal{D}_{fix}$) during SFT results in a dramatic regression in compression efficiency. On GSM8K, the average token consumption doubles ($210 \rightarrow 425$), and on MATH-500, it surges to 926 tokens. This highlights a critical failure mechanism: $\mathcal{D}_{fix}$ is indispensable for grounding the semantic meaning of the compression control tokens. Because $\mathcal{D}_{fix}$ presents the same problem instantiated across multiple compression levels ($\gamma \in \{0.2, \ldots, 1.0\}$), it forces the model to learn the differential structural constraints imposed by each ratio (e.g., distinguishing the necessary conciseness of $\gamma=0.2$ from the elaboration allowed at $\gamma=0.8$). Without this explicit **ratio-to-structure alignment**, the generation head fails to comprehend the specific constraints implied by the policy's budget commands. Consequently, even if the policy selects an aggressive budget, the model lacks the execution capability to realize it, reverting instead to its pre-trained, verbose default.

*Table 7.* **Comprehensive Ablation Study.** We evaluate the impact of removing key components from the SFT and RL training stages. **w/o Fixed SFT**: Removing the Fixed-Ratio Cohort ($\mathcal{D}_{fix}$) in Stage 2. **w/o $\mathcal{R}_{\ldots}$**: Removing specific reward terms from the CHRPO objective. Metrics reported are Token Count (Tok) and Accuracy (Acc).

| Method Variant | GSM8K | | MATH-500 | | MetaMath-1k | |
|---|---|---|---|---|---|---|
| | Tok ↓ | Acc ↑ | Tok ↓ | Acc ↑ | Tok ↓ | Acc ↑ |
| **Full CHRPO** | **210** | **85.8** | 452 | **64.8** | **213** | 91.1 |
| *SFT Stage Ablation* | | | | | | |
| w/o Fixed SFT | 425 | 85.0 | 926 | 63.8 | 404 | 90.4 |
| *RL Reward Ablations* | | | | | | |
| w/o $\mathcal{R}_{cal}$ (Calib.) | 312 | 85.3 | **431** | 62.4 | 310 | 90.9 |
| w/o $\mathcal{R}_{ctrl}$ (Ctrl.) | 258 | 82.1 | 568 | 59.6 | 267 | 89.0 |
| w/o $\mathcal{R}_{mode+ctrl}$ | 324 | 85.7 | 604 | 60.0 | 334 | **91.2** |

**Impact of Budget Calibration Reward ($\mathcal{R}_{cal}$).** The Calibration Reward imposes a penalty proportional to the deviation between the selected ratio token's implied budget and the realized generation length. Ablating this component reveals **divergent instability patterns** contingent on task difficulty:

- **On easier tasks (GSM8K, MetaMath):** Without the strict length penalty, the model exhibits uncontrolled verbosity. Token usage on GSM8K increases significantly ($210 \rightarrow 312$) as the generator reverts to lazy verbose reasoning paths, effectively disregarding the policy's low-budget mandate.

- **On harder tasks (MATH-500):** The effect is inverted but equally detrimental. While token usage decreases slightly ($452 \rightarrow 431$), accuracy degrades notably ($64.8\% \rightarrow 62.4\%$). This suggests that without $\mathcal{R}_{cal}$ anchoring the generation to the chosen budget, the aggressive shortening pressure from $\mathcal{R}_{mode}$ compromises reasoning integrity. The model tends to yield structurally incoherent or fragmented chains that respect the implicit shortness preference but fail to solve complex problems.

Thus, $\mathcal{R}_{cal}$ acts as a critical stabilizer, ensuring that the generated rationale faithfully aligns with the policy's optimal budget selection across all difficulty levels.

## F. Training and Implementation Details

**SFT Hyperparameters.** We employ **full-parameter** Supervised Fine-Tuning (SFT) using the AdamW optimizer with a learning rate of $2 \times 10^{-5}$, a cosine decay schedule featuring a $0.05$ warmup ratio, and a weight decay of $0.05$. The model is trained for 3 epochs utilizing the DeepSpeed ZeRO-3 optimization stage. The global batch size is managed via a per-device batch size of 2 and a gradient accumulation steps

of 8. All training is conducted in `bfloat16` precision with gradient checkpointing enabled to optimize memory efficiency.

**RL Hyperparameters (CHRPO).** The policy model is initialized from the converged SFT checkpoint. For the RL stage, we use the AdamW optimizer with a conservative learning rate of $1 \times 10^{-6}$, 10 warmup steps, and a weight decay of $0.1$. To stabilize training, the maximum gradient norm is clipped at $1.0$. Notably, we eliminate the KL-divergence penalty term in both the reward formulation and the loss function, allowing the policy to explore the solution space without being tethered to the SFT prior distribution. The entropy coefficient is set to $0$. We employ PPO-style objective clipping with asymmetric thresholds of $0.2$ and $0.28$. The RL training spans 10 epochs on 4 GPUs. During the rollout phase, we utilize a sampling temperature of $0.7$ and a nucleus sampling ($top$-$p$) value of $0.9$. The data flow is configured with a prompt batch size of $64$, a generation batch size of $128$, and a training mini-batch size of $32$. For each prompt, we generate $G = 4$ parallel responses to estimate the advantage.

## G. Baselines and Reproducibility

**Training-Free Baselines.** We detail the implementation of the two training-free baselines used to benchmark inference-time intervention:

- **LC-Prompt (Length-Control Prompting):** We prepend an explicit length-constraint instruction to the system prompt: *"Please reduce 40% of the words in your Chain-of-Thought process."* Decoding hyperparameters remain consistent with the default setting. This baseline evaluates the model's innate ability to follow compression instructions via in-context learning (targeting $\gamma \approx 0.6$).

- **Hard Truncation:** We enforce a hard token budget by setting the generation parameter `max_new_tokens` to $0.6 \times L_{\max}$ (targeting $\gamma = 0.6$). This serves as a lower-bound baseline for blindly cutting off reasoning.

**Thinkless Re-implementation.** To ensure a rigorous and fair comparison with Thinkless, we reproduce their method while aligning the experimental variables. We map their dual-mode reasoning (Short vs. Long) to our specific compression levels: the **Short** path is trained using our $\gamma = 0.2$ compressed CoT data, and the **Long** path uses the original uncompressed ($\gamma = 1.0$) CoT. Following their official two-stage protocol, we first perform SFT to teach the model to distinguish and generate these two styles conditional on a selection token. Subsequently, we execute the DeGRPO RL stage using the exact same RL dataset $\mathcal{S}$ and base model

| Budget $\gamma$ | Ours (Wins) | llmlingua-2 (Wins) | Ours Pref. (%) |
|---|---|---|---|
| 0.2 | 49.4 | 0.6 | 98.8 |
| 0.4 | 49.8 | 0.2 | 100.0 |
| 0.6 | 47.6 | 2.4 | 95.2 |
| 0.8 | 42.0 | 8.0 | 84.0 |

*Table 8.* Human pairwise preferences comparing **Ours** vs. **llmlingua-2** (5 raters $\times$ 50 prompts = 250 votes per budget). "Wins" columns are the average counts (out of 50) per rater.

backbone as our Extra-CoT method. By keeping the core decoding strategies and optimization hyperparameters identical, this re-implementation isolates the algorithmic effectiveness of the policy optimization method.

## H. User Study Design and Results

**Protocol.** We conducted a blind A/B preference study involving **five expert raters** to evaluate the qualitative quality of the compressed reasoning chains. We compared our method (Extra-CoT) against the baseline, llmlingua-2, across four distinct target budgets $\gamma \in \{0.2, 0.4, 0.6, 0.8\}$. Each rater evaluated $50$ randomly sampled prompts per budget in a side-by-side comparison setup. This process resulted in a total of **250 preference votes** per budget tier ($1,000$ votes in total).

**Results.** Table 8 summarizes the mean win counts and the pairwise preference rate (computed over non-ties). The results demonstrate a decisive advantage for our method:

- **Dominance in Extreme Compression:** In the high-compression regime ($\gamma \le 0.4$), our method achieved a near-total preference rate ($\ge 98.8\%$). Even at the most aggressive $\gamma = 0.2$ budget (80% reduction), human raters almost unanimously favored Extra-CoT. This strongly supports our hypothesis that semantically-preserved selection maintains reasoning coherence significantly better than the baseline's perplexity-based pruning.

- **Alignment with Automatic Evaluation:** The preference gap is widest at low budgets and narrows slightly at the mildest compression ($\gamma = 0.8$, 84.0%), perfectly mirroring the trend observed in the GPT-4o automatic judgments reported in the main text. This consistency validates the use of GPT-4o as a reliable proxy for human preference in this task.

