# OpenReview forum: "Towards Efficient Large Language Reasoning Models via Extreme-Ratio Chain-of-Thought Compression"
_ICML.cc/2026/Conference — ICML 2026 regular_

### Official Review · Reviewer_uvHx · 2026-03-11

**Soundness:** 3
**Presentation:** 3
**Significance:** 3
**Originality:** 3
**Overall Recommendation:** 4
**Confidence:** 4

**Summary:**

The paper “Towards Efficient Large Language Reasoning Models via Extreme-Ratio Chain-of-Thought Compression” addresses a practical problem in reasoning LLMs: chain-of-thought (CoT) improves accuracy but greatly increases inference cost. The authors argue that existing CoT compression methods break down at very high compression levels because they lose logical fidelity. To solve this, they propose Extra-CoT, a three-stage framework for preserving answer quality while aggressively shrinking reasoning traces.
First, the paper trains a semantically preserved, question-aware compressor for mathematical CoT data. This compressor uses fine-grained, formula-aware annotations so that important equations and reasoning steps are retained instead of being fragmented or dropped. Second, the compressed reasoning data are used in a mixed-ratio supervised fine-tuning stage, where the model learns to follow different compression budgets, such as 20%, 40%, 60%, 80%, and 100%. This teaches controllable compression and prepares the model for adaptive budget selection. Third, the authors introduce CHRPO (Constrained and Hierarchical Ratio Policy Optimization), a reinforcement-learning stage that learns to choose lower reasoning budgets when possible while preserving accuracy. Experiments on GSM8K, MATH-500, and AMC2023 show that Extra-CoT is especially effective in the extreme low-ratio regime, where prior methods degrade sharply. On MATH-500 with Qwen3-1.7B, the method achieves about 73% token reduction while slightly improving accuracy over the baseline. Additional results on SVAMP, MultiArith, MetaMath-1k, and MMLU STEM suggest the method remains more robust than competing approaches under aggressive compression.

**Compliance With Llm Reviewing Policy:**

Affirmed.

**Final Justification:**

I will still go with my earlier decision weak accept.

**Key Questions For Authors:**

This is a promising and well-executed paper with a clear contribution, especially for extreme-ratio compression in mathematical reasoning. Its strongest value lies in showing that carefully designed supervision plus adaptive optimization can preserve accuracy under severe token constraints. The main limitation is that the evidence currently supports a domain-specific success story more strongly than a broad claim about efficient reasoning in general.

Following are some of the suggestions to improve it further:


The abstract should mention that the method is evaluated primarily on mathematical reasoning tasks and that the compressor is explicitly designed to preserve mathematical structure.

Soften the strongest causal language in the introduction unless the authors add experiments directly isolating fidelity from the rest of the pipeline.

Add a compact comparison table summarizing differences in supervision source, controllability, budget adaptivity, and whether methods are task-agnostic or domain-specialized.

Briefly discuss the tradeoff between extractive fidelity and abstractive succinctness, especially under extreme ratios.

Clarify the domain assumptions and add discussion of how the compressor would transfer to non-math reasoning domains.

Add a comparison against simpler policy warm-up strategies, such as uniform ratio assignment or random difficulty grouping.

Replace “guarantee” with more cautious wording unless a formal argument is provided. Also discuss how much RL data is filtered out by the monotonicity criterion and whether this introduces selection bias.

Report multi-seed results or at least confidence intervals on the main benchmarks, especially for the strongest claims at extreme compression.

Reframe the main claim more precisely: the strongest contribution is preserving reasoning accuracy under aggressive compression, rather than uniformly improving all operating points.

**Limitations:**

Add inter-rater agreement statistics and, if possible, a small human study on end-to-end model outputs rather than only compressed rationales.

Add inter-rater agreement statistics and, if possible, a small human study on end-to-end model outputs rather than only compressed rationales.

**Strengths And Weaknesses:**

Strengths:

Tackles a timely and important problem: reducing chain-of-thought inference cost while trying to preserve reasoning accuracy under aggressive compression. The paper explicitly motivates the inefficiency of verbose CoT and targets the extreme-compression regime where prior methods degrade sharply.

Proposes a clear, coherent three-stage framework called Extra-CoT (i) a semantically preserved, question-aware compressor,(ii) mixed-ratio supervised fine-tuning, and (iii)
CHRPO for RL-based budget optimization.

The compressor is task-aware and mathematically informed, using question attention and formula-aware annotations to preserve important reasoning units and mathematical integrity.

Strong emphasis on ratio controllability: the mixed-ratio SFT stage is designed to improve adherence to requested compression budgets, which is a practical strength over baselines.

The RL stage, CHRPO, is a meaningful methodological addition that explicitly incentivizes lower budgets while preserving accuracy.

Reports strong results in the hardest low-ratio regime, especially against TokenSkip and Thinkless. The paper claims over 73% token reduction on MATH-500 with a slight accuracy improvement, and shows robust behavior at a 0.2 compression ratio where baselines collapse.

Weaknesses:

The evaluation is narrowly centered on mathematical reasoning, so the evidence does not yet support broad claims about efficient reasoning in general. The main benchmarks highlighted are GSM8K, MATH-500, and AMC2023.

The proposed compressor is domain-specialized, not generic: it relies on formula-aware, math-preserving supervision, which may limit transfer to non-math domains.

Some of the method’s gains appear to come not only from the learning algorithm, but also from a stronger supervision pipeline built with task-specific annotations.

The paper relies partly on LLM-based judging for compressor quality, which is helpful but weaker than broader human evaluation. The human evaluation is only mentioned as supporting evidence.

The claims around CHRPO can sound stronger than the evidence warrants; the ablations support usefulness, but not necessarily formal “guarantee”-style interpretations. The paper mainly shows empirical benefit.

Evidence for general robustness beyond the math setting remains limited, even though the paper notes gains in constrained-context settings and points to appendices for more results.

---

> ### Author Rebuttal · Authors · 2026-03-30
>
> We sincerely thank you for recognizing our framework's design, ratio controllability, and extreme low-ratio performance. Your constructive feedback significantly improves our paper's rigor. We address your concerns below.
>
> ### Q1. Narrow scope, causality claims, transferability, and extractive vs. abstractive trade-offs
>
> We agree our formula-aware compressor is currently **math-specialized.** We will revise the abstract/intro to explicitly state this and soften causal language to reflect that gains stem from **a combined pipeline effect**. While our SFT and CHRPO stages are domain-agnostic, transferring to domains like coding requires code-specific supervision.
>
> Regarding extreme ratio trade-offs, we intentionally prioritize **extractive fidelity** over abstractive fluency. Preserving mathematical symbolic anchors is far more critical than fluent natural language, as abstractive rewriting under strict budgets risks paraphrastic drift and structural damage.
>
> ### Q2. Source of gains, inter-rater agreement, and end-to-end evaluation
>
> To isolate the exact source of our gains and assess end-to-end performance, we conducted comprehensive human evaluations against two strong baselines.
>
> **1. Isolating Learning Algorithm Gains (vs. LLMLingua-2)**
>
> To ensure gains are not solely from supervision strength, we tested a matched control: LLMLingua-2 trained on our identical GPT-4o dataset. Evaluators consistently preferred our compressor, confirming our learning algorithm provides significant independent gains.
>
> | Budget γ | Ours Pref. (%) |
> | -------- | -------------- |
> | 0.2      | 97.6           |
> | 0.4      | 95.2           |
> | 0.6      | 90.8           |
> | 0.8      | 82.4           |
>
> **2. End-to-End Evaluation & Agreement (vs. TokenSkip)**
>
> To verify benefits on final answers, we evaluated end-to-end outputs and calculated Fleiss’ κ. Extra-CoT is strongly preferred across all budgets with high inter-rater agreement, supporting robustness beyond the rationale level.
>
> | Budget γ | Win % | Lose % | Fleiss’ κ |
> | -------- | ----- | ------ | --------- |
> | 0.2      | 97.5% | 2.5%   | 0.93      |
> | 0.4      | 78.0% | 22.0%  | 0.91      |
> | 0.6      | 85.0% | 15.0%  | 0.84      |
> | 0.8      | 81.5% | 18.5%  | 0.86      |
>
> ### Q3. Summarize methodological differences
>
> We will include the following comparison table in the revision to clarify our contributions:
>
> | Method               | Supervision Source          | Controllability | Budget Adaptivity | Domain           |
> | -------------------- | --------------------------- | --------------- | ----------------- | ---------------- |
> | **LLMLingua-2**      | Generic distillation        | No              | No                | Task-agnostic    |
> | **TokenSkip**        | Compressor-based            | Fixed ratio     | No                | Mostly agnostic  |
> | **Thinkless**        | Short/long supervision + RL | Limited         | Yes               | Agnostic         |
> | **Extra-CoT (Ours)** | Formula-aware + Mixed-ratio | Fixed + Policy  | Yes               | Math-specialized |
>
> ### Q4. Compare against simpler policy warm-up strategies
>
> We added an SFT-stage ablation comparing our method against a uniform/random grouping strategy. As shown below, our difficulty-aware warm-up consistently outperforms the baseline. Importantly, it achieves **higher accuracy with lower or comparable ActRatios,** confirming that aligning reasoning burden with budget yields a superior policy warm-up.
>
> | **Warm-up Strategy**        | **GSM8K Acc@all** | **GSM8K ActRatio** | **MATH Acc@all** | **MATH ActRatio** | **MetaMath Acc@all** | **MetaMath ActRatio** |
> | --------------------------- | ----------------- | ------------------ | ---------------- | ----------------- | -------------------- | --------------------- |
> | Uniform&Random grouping     | 84.6              | 0.74               | 56.4             | 0.79              | 88.0                 | 0.76                  |
> | **Difficulty-aware (Ours)** | 86.3              | 0.65               | 63.0             | 0.78              | 91.6                 | 0.69                  |
>
> ### Q5. Cautious wording for "guarantees" and discussing filtered data
>
> We will replace "guarantee" with **"empirical behavioral properties"** or **"risk-sensitive shaping effects."** Regarding the monotonicity criterion, we will clarify that it is a selective data-construction step designed for high-precision teacher budgets, retaining ~91.6% of the RL pool.
>
> ### Q6. Report multi-seed results for extreme compression claims
>
> We will report multi-seed results (over 5 seeds) for key benchmarks using the post-CHRPO Extra-CoT model (Qwen3-1.7B). Results are highly consistent across random seeds, preserving our main trends. The slight AMC23 variation is expected given its smaller evaluation set.
>
> | Dataset      | Mean ± Std Dev   |
> | ------------ | ---------------- |
> | **GSM8K**    | **85.81 ± 0.22** |
> | **MATH-500** | **64.76 ± 0.43** |
> | **AMC23**    | **50.50 ± 1.12** |

---

> > ### Author Rebuttal · Reviewer_uvHx · 2026-04-02
> >
> > I am satisfied with the author replies to my queries.

---

> > > ### Author Response · Authors · 2026-04-03
> > >
> > > Dear Reviewer uvHx,
> > >
> > > Thank you again for your invaluable time and effort on our paper. Thank you very much for approving our work and rebuttal!
> > >
> > > Sincerely yours,
> > >
> > > The Authors

---

### Official Review · Reviewer_4WLX · 2026-03-12

**Soundness:** 2
**Presentation:** 3
**Significance:** 2
**Originality:** 3
**Overall Recommendation:** 4
**Confidence:** 4

**Summary:**

This paper introduce a comprehensive multi-stage CoT compression framework named Extra-CoT. In all the steps, some novel heuristic-based techniques are proposed. The overall performance show that the whole framework achieves adaptive compression ratio of CoT while maintains the accuracy on Mathematical benchmarks.

**Compliance With Llm Reviewing Policy:**

Affirmed.

**Final Justification:**

After the rebuttal, most of my concerns are addressed. I will raise my score. Now I tend to recommend acceptance for this paper.

**Key Questions For Authors:**

See above.

**Limitations:**

Yes

**Strengths And Weaknesses:**

Strength:
1. The framework consisting of three stages (compressor training, mixed-ration SFT and RL with hierarchical reward) is novel.
2. The empirical performance of the proposed method is impressive. On some benchmarks, Extra-CoT even outperforms baselines on both tokens consumption and accuracy.
3. The presentation is very clear and precise. The method proposed by this paper is very complicated and heuristic, making a detailed elaboration difficult.

Weakness:
1. The method is too heuristic. Some part seems unreasonable and lack of theoretical or insightful explanations. For example:
- Lines 185-195 introduces the annotation of importance score. Why can GPT-4o accomplish this goal? Judging the importance of each unit in a sentence may not be a standard and easy task for any LLM, not to mention the importance is question and answer-aware.
- The final compression label $\tilde z_\gamma$ is defined as the concatenation of the kept tokens. However, in almost all examples, the concatenation of most important tokens selected by GPT-4o may not be complete and decent sentences. Why using this incomplete sentences as the label of compressed CoT?
- The heuristic-based difficulty-selective model in Appendix C seems to be out of no where. What is the theoretical or empirical basis of using the four dimension to judge the difficulty of CoT?
2. There are too many hyper-parameters in the proposed method. For example, only in the CHRPO stage, there are $\lambda_{short}, \lambda_{over}, \lambda_{under}^{err}, \lambda_{over}^{err}, \epsilon, \lambda_{\varnothing}, r_0, \eta_{short}, \eta_{over}, \eta_{under}^{err}, \eta_{over}^{err}, w_{acc} and w_{cal}$. There is no sufficient sensitivity analysis to these hyper-parameters, and no guidance for setting this hyper-parameters in practice.
3. The ablation study is not sufficient. For example, $\mathcal{R}_{\varnothing}$ and the clip operation in Eq. (11) are not ablated.

---

> ### Author Rebuttal · Authors · 2026-03-30
>
> We sincerely thank you for recognizing our novelty, performance, and clarity. Below we address your helpful feedback to further strengthen the paper.
>
> ### Q1. The Heuristic Nature of the Method
>
> **Concern:** Several heuristic components lack theoretical explanations.
>
> While heuristic, these components provide a lightweight, effective recipe for extreme-ratio compression. These choices are empirically backed:
>
> - **Five-ratio curriculum:** Outperforms an endpoints-only approach (Appendix C), justifying the need for dense semantic anchors.
> - **Difficulty-selective warm-up:** Initializes the SFT `<COMP_POLICY>` cohort under controllable budgets, acting as a practical warm-start rather than a theoretical estimator.
> - **Monotonic filtering:** Empirically helpful to filter out lucky guesses and ensure stable RL supervision.
>
> We will clarify these as motivated operational choices in the revision.
>
> ### Q2. Basis of the Difficulty-Selective Model
>
> **Concern:** What is the basis for using the four dimensions in Appendix C?
>
> These dimensions are compute-free proxies for reasoning burden used to initialize the `<COMP_POLICY>` SFT warm-up. Specifically, length approximates overall reasoning load; equation/step count reflects procedural complexity; and operator/lexical richness capture symbolic and conceptual diversity. These are computed directly from the gold CoT, keeping the warm-up practical.
>
> Empirically, this structured heuristic outperforms random grouping, improving GSM8K accuracy (84.6 $\rightarrow$ 86.3, lower ActRatio 0.74 $\rightarrow$ 0.65), MATH (56.4 $\rightarrow$ 63.0, similar ActRatio (0.79 vs. 0.78)), and MetaMath (88.0 $\rightarrow$ 91.6, lower ActRatio 0.76 $\rightarrow$ 0.69).
>
>
> ### Q3. GPT-4o's Annotation Capability
>
> **Concern:** Why can GPT-4o accomplish question/answer-aware token importance scoring?
>
> GPT-4o performs **structured span-selection** of an indexed CoT rather than open-ended scoring, making supervision verifiable.
>
> To address reliability, we conducted a **human validity check** on the compressed CoTs generated by GPT-4o. Our human validity check on GPT-4o's outputs confirms high performance across three criteria. These results suggest that GPT-4o can serve as a highly effective annotator under this constrained protocol.
>
> | Metric                | Yes % | No %  |
> | --------------------- | ----- | ----- |
> | Math Fidelity         | 93.6% | 6.4%  |
> | Reasoning Coherence   | 88.8% | 11.2% |
> | Clarity & Readability | 84.8% | 15.2% |
>
> ### Q4. Incomplete Sentences as Labels
>
> **Concern:** Concatenated tokens form incomplete sentences. Why use these as labels?
>
> Extreme-ratio compression (e.g., 20% budget) aims to distill critical mathematical anchors, not generate fluent summaries. Under strict budgets, grammatical incompleteness is an expected trade-off to preserve vital formulas and logical leaps, prioritizing information density over natural language fluency. We will clarify this trade-off in the revised version.
>
> ### Q5. Hyperparameter Sensitivity and Missing Ablations
>
> **Concern:** Too many hyperparameters without sensitivity analysis, and missing ablations.
>
> We appreciate you pointing this out. We agree the manuscript needs clearer practical guidance on hyperparameters and additional ablations.
>
> 1. **Practical Guidance on Hyperparameters:** The CHRPO coefficients are not a set of many independent free knobs; they form a small number of functional groups with distinct roles. Specifically, the $\lambda$-terms shape response-level compression, the $\eta$-terms with $r_0$ guide first-token ratio selection, and $w_{\text{acc}}$, $w_{\text{cal}}$, $\epsilon$, and $\lambda_{\emptyset}$ govern accuracy balancing, budget calibration, and the explicit `<think>` constraint. The key principle is to preserve their relative ordering—for example, the penalty for over-aggressive shortening that causes failure should be stronger than that for conservative over-budget behavior. We will clarify this grouping and these setting principles in the revised version to improve clarity and reproducibility.
> 2. **Missing Ablations:** Removing $R_{\emptyset}$ yields abnormally short, less accurate outputs, confirming it prevents bypassing explicit reasoning. Removing the outer $clip$ mildly degrades the token-accuracy trade-off, showing it acts as a reward stabilizer. We will expand Table 5:
>
> | Method Variant | GSM8K Tok ↓ | GSM8K Acc ↑ | MATH-500 Tok ↓ | MATH-500 Acc ↑ | MetaMath-1k Tok ↓ | MetaMath-1k Acc ↑ |
> | -------------- | ----------- | ----------- | -------------- | -------------- | ----------------- | ----------------- |
> | **w/o R∅**     | 195         | 82.8        | 420            | 62.2           | 203               | 88.8              |
> | **w/o clip**   | 237         | 85.0        | 478            | 63.4           | 241               | 90.1              |
> | Full CHRPO     | 210         | 85.8        | 452            | 64.8           | 213               | 91.1              |

---

> > ### Author Rebuttal · Reviewer_4WLX · 2026-04-03
> >
> > Even with the key principle to preserve their relative ordering, the tuning of hyper-parameters is still too hard. The importance order is heuristically decided, and after all the hyperparameter space is still large.

---

> > > ### Author Response · Authors · 2026-04-05
> > >
> > > Dear Reviewer 4WLX,
> > >
> > > Thank you again for your time and careful reading of our paper. We also sincerely appreciate your positive assessment of our work and rebuttal.
> > >
> > > We agree that our previous response did not fully resolve your concern. In the original CHRPO formulation, the reward is indeed written with multiple explicit coefficients, and the relative ordering among the case-wise rewards and penalties still reflects a heuristic design choice. So our claim is **not** that CHRPO is free of heuristic constants. Rather, our point is that its **practical tuning burden is much lower-dimensional than the raw number of coefficients may suggest**.
> > >
> > > To examine this more directly, we implemented a reduced **2-group** version of CHRPO. Here, “tying” does **not** mean forcing all coefficients to be the same. Instead, it means keeping the internal relative structure of each reward family unchanged, while replacing several independently tunable coefficients with a single family-level scale. Concretely, all case-wise coefficients in $\(R_{\text{mode}}\)$ are scaled by one factor $\(\alpha_{\text{mode}}\)$, and all case-wise coefficients in $\(R_{\text{ctrl}}\)$ are scaled by another factor $\(\alpha_{\text{ctrl}}\)$. The remaining constants (such as $\(\kappa\)$, $\(\epsilon\)$) are fixed globally at the default CHRPO setting and are not retuned for each dataset.
> > >
> > > Under this reduced parameterization, the default 2-group setting $\((\alpha_{\text{mode}}$, $\alpha_{\text{ctrl}})=(1.0, 1.0)\)$ still achieves **85.6 / 223** on GSM8K, **64.5 / 462** on MATH-500, and **90.9 / 226** on MetaMath. These are very close to the full CHRPO results reported in the paper: **85.8 / 210**, **64.8 / 452**, and **91.1 / 213**, respectively. In other words, reducing the reward from many explicit coefficients to two family-level scales leads to only a very small degradation relative to full CHRPO.
> > >
> > > We also ran coarse local perturbations around this reduced setting. With \((0.8, 1.0)\), the model obtains **85.2 / 224** on GSM8K, **63.7 / 453** on MATH-500, and **90.6 / 217** on MetaMath. With \((1.0, 0.8)\), it obtains **85.4 / 228**, **64.2 / 472**, and **90.7 / 233**, respectively. These changes are small and smooth: the accuracy varies only within **0.2–0.8 points** of the reduced default across the three benchmarks, and token usage changes only modestly. This does not look like a system that requires fine-grained retuning over a large effective hyperparameter space.
> > >
> > > So we agree that a heuristic prior still remains in the fixed case structure and relative ordering. But this experiment suggests that, once that structure is fixed globally, the **main practical tuning freedom is effectively concentrated in two family-level scales**, rather than in a large set of independently sensitive coefficients. We will revise the paper to make this distinction clearer: CHRPO is **not heuristic-free**, but its **effective tuning dimension is substantially smaller than the raw coefficient count might imply**.
> > >
> > > Sincerely yours,
> > > The Authors

---

### Official Review · Reviewer_eATX · 2026-03-12

**Soundness:** 3
**Presentation:** 3
**Significance:** 3
**Originality:** 3
**Overall Recommendation:** 4
**Confidence:** 4

**Summary:**

This paper presents a method for chain-of-thought compression consisting of three main stages:

Semantic-preserving CoT compression: This component generates compressed CoT data that they use for training a compressor model. The main improvement from previous work is ensuring the integrity of formulas/symbolic expressions during compression.

Mixed-ratio SFT: The model is finetuned on CoTs of various compression lengths. A control token is introduced into the model, so that a “COMP_X” token precedes a reasoning trace that has been compressed by a ratio of X. A special “COMP_POLICY” token is also introduced to allow the model to choose its own compression ratio in the subsequent RL stage.

Reinforcement learning: the reinforcement learning stage uses a custom reward with separate components for the reasoning model and control head. The main reward is designed to conservatively shorten CoTs (only when correct), avoid collapse (encourage longer when wrong), and adhere to the compression ratio. The control head reward helps learn compression budgets that match the teacher budget (smallest ratio that still gives correct answers).

Main results: on GSM8K, Math 500, and AMC2023, their method achieves stronger length-accuracy trade-off than the base model, TokenSkip, and Thinkless.
Additional analyses are provided on the quality of compressed CoTs and an ablation of reward components.

**Compliance With Llm Reviewing Policy:**

Affirmed.

**Final Justification:**

See rebuttal acknowledgement. The proposed method is sound, extensively ablated, and achieves strong empirical results. My decision is limited to a weak accept because the method currently relies on a domain-specific intervention (preserving mathematical formulas) while its applicability (specifically that of CHRPO) to other domains is not explored.

**Key Questions For Authors:**

See weaknesses.

- How does the post-CHRPO model perform when using user-specified compression ratios? Providing these results could help quantify how much of its performance comes from selecting good compression ratios vs. more accurate reasoning chains under a given compression ratio.

**Limitations:**

yes

**Strengths And Weaknesses:**

Strengths:
- They present strong empirical improvements over baselines at extreme compression ratios. Their method seems able to compress reasoning chains more than baselines without loss of accuracy, which is a significant result.
- The analysis and ablations section is thorough and isolates the contribution from each component.

Weaknesses
- Comparison to LLMLingua-2: It is not clear whether the quality improvement comes entirely from their method to preserve mathematical formulas. The natural text from their compressor seems notably more coherent than from LLMLingua-2. Does this indicate a difference in the supervision data (theirs from GPT-4o)?
- Benchmarks: GSM8k, MATH500, and AMC2023 are relatively old and potentially suffer from data contamination. The results would be more convincing if evaluated on more recent benchmarks (e.g., AMC25).
- Minor: the naming of several components seem exaggerated. For example, “hierarchical” seems inaccurate for describing the reward, and “rationale integrity” only refers to a format reward for <think> tags here. I believe the presentation would be clearer if such names/titles were exclusive to core components/contributions.

---

> ### Author Rebuttal · Authors · 2026-03-30
>
> We sincerely thank you for the constructive feedback. Below, we address your four main concerns regarding the source of our gains, benchmark recency, terminology, and CHRPO's impact on fixed-ratio reasoning.
>
> ### Q1. Comparison to LLMLingua-2 & Source of Gains
>
> **Concern:** Is the quality improvement driven entirely by the proposed method (preserving formulas) or simply by using GPT-4o for supervision?
>
> We agree that this is a critical distinction.
> Our improvement stems from our question-aware, formula-preserving annotation strategy, not just the GPT-4o teacher. To disentangle teacher strength from the compression mechanism, we conducted a matched control experiment.
>
> We replaced LLMLingua-2’s original supervision with GPT-4o, trained its standard sliding-window compressor, and compared this GPT-4o-enhanced LLMLingua-2 against our Extra-CoT compressor. Five human evaluators compared the outputs. As shown below, our compressor is consistently preferred across all budgets. This confirms our advantage comes from the proposed explicit preservation of mathematical structure, not just teacher quality.
>
> | Budget γ | Ours Pref. (%) |
> | -------- | -------------- |
> | 0.2      | 97.6           |
> | 0.4      | 95.2           |
> | 0.6      | 90.8           |
> | 0.8      | 82.4           |
>
> ### Q2. Benchmarks and Contamination Risks
>
> **Concern:** GSM8K, MATH-500, and AMC2023 are relatively old and may suffer from data contamination. Results on recent benchmarks like AMC25 would be more convincing.
>
> We acknowledge that older benchmarks risk data contamination, potentially inflating absolute accuracy. Therefore, we primarily use GSM8K, MATH-500, and AMC2023 for matched-model relative comparisons, as all methods share the same backbone and evaluation setup.
>
> To address the recency concern more directly, we additionally evaluate on AMC25 under `<COMP_POLICY>`. Extra-CoT continues to achieve the best accuracy-compression trade-off there, which strengthens our claim that the observed advantage is not confined to older benchmarks alone.
>
> | Method     | Acc@all | ActRatio |
> | ---------- | ------- | -------- |
> | Qwen3-1.7B | 47.5    | -        |
> | TokenSkip  | 17.5    | 0.57     |
> | Extra-CoT  | 45.0    | 0.26     |
>
> ### Q3. Overstated Naming
>
> **Concern:** Terms like "hierarchical" and "rationale integrity" seem exaggerated for describing the reward components.
>
> We fully accept this critique; these terms should be more restrained and precise.
>
> - We will replace "hierarchical" with **"temporally decomposed reward,"** accurately reflecting that the main reward applies to the full rationale while the control reward targets the first ratio-selection token.
> - We will describe $R_{\emptyset}$ more conservatively as a **format-constraint reward**, as it simply prevents the policy from bypassing the budget by omitting the `<think>` block.
> - We will thoroughly review and tone down similar terminology in the revision to improve clarity and academic rigor.
>
> ### Q4. Post-CHRPO Performance on User-Specified Ratios
>
> **Concern:** How does the post-CHRPO model perform using user-specified compression ratios? Does performance come from selecting good ratios or generating better chains under a fixed ratio?
>
> We agree that this distinction is important. In our framework, CHRPO improves both **which budget is chosen** and **how well the model executes compressed reasoning once a budget is given**. As described in our method, the **control-head reward** targets the first token (ratio selection), while the **main reward** applies to the full rationale, updating the shared generation backbone.
>
> To demonstrate this, we report the **post-CHRPO model under externally specified fixed ratios** below, complementing the SFT fixed-ratio results already reported in the main paper. The model remains highly capable across all fixed budgets, proving that CHRPO improves not only the adaptive budget selection through `<COMP_POLICY>`, but also the execution quality of compressed reasoning under user-specified fixed ratios.
>
> | Methods           | Ratio | GSM8K ActRatio | GSM8K Acc@all | MATH-500  ActRatio | MATH-500 Acc@all | MetaMath ActRatio | MetaMath Acc@all |
> | ----------------- | ----- | -------------- | ------------- | ------------------ | ---------------- | ----------------- | ---------------- |
> | Extra-CoT (CHRPO) | 0.8   | 0.75           | 88.5          | 0.76               | 67.0             | 0.73              | 93.2             |
> | Extra-CoT (CHRPO) | 0.6   | 0.58           | 88.3          | 0.58               | 66.0             | 0.54              | 92.8             |
> | Extra-CoT (CHRPO) | 0.4   | 0.39           | 87.9          | 0.38               | 63.0             | 0.39              | 92.6             |
> | Extra-CoT (CHRPO) | 0.2   | 0.25           | 86.0          | 0.23               | 60.8             | 0.27              | 91.7             |

---

> > ### Author Rebuttal · Reviewer_eATX · 2026-04-02
> >
> > I am satisfied with the authors' responses to my questions and concerns. After reading the responses from other reviewers, I maintain my decision of weak accept. The primary factors preventing a strong accept is that the core technique -- preserving the integrity of mathematical formulas -- is relatively straightforward and restricted to the math domain. That said, the empirical results are solid, and the RL component (CHRPO) does seem possible to apply to other domains (though this was not explored in the paper).

---

> > > ### Author Response · Authors · 2026-04-03
> > >
> > > Dear Reviewer eATX,
> > >
> > > Thank you again for your invaluable time and effort on our paper. We also greatly appreciate your suggestion regarding extension beyond the current math setting.
> > >
> > > We agree that the current compressor is math-specialized, and that our strongest direct evidence is in mathematical reasoning. In the current paper, the specialization mainly comes from the supervision design: we explicitly preserve mathematical formulas and symbolic anchors because these are the most fragile and decision-critical parts under extreme compression.
> > >
> > > That said, we do not view the underlying compression principle as limited to math. More broadly, our key idea is domain-aware, structure-preserving compression: identify and preserve the spans whose structural integrity is most critical for correct reasoning under a strict token budget. Under this view, a natural extension to other domains would be to replace formula-aware supervision with the corresponding domain-specific structural anchors.
> > >
> > > For example, in coding, this would mean preserving syntax- and dependency-critical spans, such as function signatures, control-flow conditions, variable definitions and updates, API calls, and error-handling branches. In scientific reasoning, the analogous anchors would include equations, units, variable/entity definitions, causal statements, key assumptions, and experimental settings or result statements. In more general reasoning settings, the preserved structure could focus on constraints, premises, quantities, evidence-bearing spans, and conclusion-critical relations.
> > >
> > > Therefore, we do not claim that the current compressor is already task-agnostic. Rather, we view this work as evidence that structure-aware compression can be transferred beyond math by redefining what counts as the domain’s critical structure, with coding, scientific reasoning, and broader general reasoning as natural next directions. We will continue exploring these broader applications in future work.
> > >
> > > Sincerely yours,
> > >
> > > The Authors

---

### Official Review · Reviewer_odPM · 2026-03-13

**Soundness:** 3
**Presentation:** 2
**Significance:** 3
**Originality:** 3
**Overall Recommendation:** 4
**Confidence:** 3

**Summary:**

Authors propose a novel method for extreme-ratio CoT.
The steps involve:
1. Semantic compressor, which uses Qwen CoTs generated when using the CAMEL dataset. Then, GPT-4o is used to select indices, and labels are assigned to tokens based on whether they were kept or not. Next, Longformer-large-4096 is trained to be the CoT compressor with Focal Loss to predict which tokens are unnecessary to keep.

2. Mixed-ratio SFT to instill controllability: Uses CoT compressor to build SFT dataset comprising two distinct cohorts. This teaches the model how to follow compression rules and also teaches the model how to predict how much to compress.
3. CHRPO optimizes the autonomous policy: Now in RL mode, the model needs to both predict the target ratio and generate a response.

The paper provides experiments comparing various methods on various benchmarks, showing increased performance while using fewer output tokens.

**Compliance With Llm Reviewing Policy:**

Affirmed.

**Final Justification:**

Based on my original comments, our discussion, and other reviews, I maintain my current score.

**Key Questions For Authors:**

1. Can you expand on the definition of realized ratio (line 252-3)?

**Limitations:**

Yes.

**Strengths And Weaknesses:**

Strengths.
1. Empirically, this method performs very well with respect to the various baselines both in terms of output tokens and accuracy, although it would be great if authors can clarify what exactly acc@all is measuring.

2. Authors compare with many different baselines, on three different datasets. Examples are provided of different outputs. Overall, the authors appear to have been rather thorough in their evaluation of the method.

Weaknesses.
1. There is little discussion about the extra level of compute needed. In particular, due to the various steps of this process, it appears as though a significant amount of additional compute may be necessary. While the final result performs well, it would be nice to discuss the amount of additional compute that is needed to achieve this result. In addition to the various training loops, it seems like data like finding the teacher budget $r^*$ for RL (as detailed in lines 269-270) would be rather expensive.

2. The overall methodology (as described in Section 3) is very difficult to follow and would significantly benefit from improved presentation. Perhaps a pseudo-code environment would better explain the steps. There are many different components to the methodology, all which require a specific training loop.

3. The rewards in the RL objective are not very clearly defined or motivated. Although Table 3 performs an ablation, it would help to better justify their necessity by motivating in writing.

---

> ### Author Rebuttal · Authors · 2026-03-30
>
> We thank the reviewer for recognizing our strong empirical performance, comprehensive baselines, and thorough evaluation. We appreciate the constructive feedback on presentation and compute overhead, which we will address in the revision.
>
> ### Q1. Definition of "Acc@all"
>
> **Concern:** Clarify what exactly Acc@all is measuring.
>
> **Acc@all** is the **accuracy computed over the entire test set**. We will clarify this explicitly.
>
> ### Q2. Additional Compute Overhead
>
> **Concern:** Little discussion about extra compute, and finding the teacher budget $r^*$ seems expensive.
>
> We will explicitly state the offline costs of our multi-stage pipeline. The four stages (**Compressor, Mixed-ratio SFT, Teacher-budget sweep, and CHRPO**) are **offline post-training / data-construction costs**, distinct from the **inference-time latency** in Table 7. The overall compute overhead is manageable in our setup. The total training pipeline takes **<2 days** with 8x NVIDIA RTX A6000 (48GB) GPUs.
>
> | Stage                | Cost |
> | -------------------- | ---- |
> | Compressor           | 5+h  |
> | Mixed-ratio SFT      | 10+h |
> | Teacher-budget sweep | 5+h  |
> | CHRPO                | 12+h |
> | Total                | 30+h |
>
> The cost of finding the teacher budget $r^*$: This step is exclusively performed offline during RL data preparation. It only requires SFT model inference across the 5 discrete target ratios ($r \in \{0.2, 0.4, 0.6, 0.8, 1.0\}$). As a pure inference pass, it is highly parallelizable and relatively inexpensive compared to RL training loops. We will detail these metrics in a "Computational Cost" paragraph in the appendix.
>
> ### Q3. Overall Methodology and Pipeline Pseudocode
>
> **Concern:** The methodology is difficult to follow and would benefit from improved presentation/pseudocode.
>
> The Extra-CoT framework operates sequentially to bridge the gap between full reasoning and extreme compression. To clarify section 3, we will add the following algorithmic overview and pseudocode:
>
> 1. **Stage 1: Semantic Compressor:** Trains an extractive compressor to learn *what* to keep, using formula-aware annotations to preserve mathematical logic.
> 2. **Stage 2: Mixed-Ratio SFT:** Teaches the LLM *how to execute* compression commands (e.g., `<COMP_40>`), preventing "control collapse". It introduces the `<COMP_POLICY>` token to warm up autonomous ratio selection.
> 3. **Stage 3: CHRPO (RL):** Optimizes *which ratio to choose*. It focuses on `<COMP_POLICY>`, rewarding the model for autonomously choosing the lowest budget that yields a correct answer.
>
> **Pseudocode for Extra-CoT Pipeline:**
>
> ```
> Algorithm: Extra-CoT Training Pipeline
> Input: math CoT data D, ratio grid R={0.2,0.4,0.6,0.8,1.0}, base model π0
>
> 1. Build span-level keep labels on D with formula-aware indexing; train compressor C.
> 2. Use C to generate compressed CoTs at ratios in R, then construct:
>    (a) fixed-ratio SFT data, and
>    (b) <COMP_POLICY> warm-up data.
> 3. Fine-tune π0 on the mixed-ratio SFT data to obtain πSFT.
> 4. For each RL training sample, run πSFT over all ratios in R;
> assign teacher budget r* as the smallest monotonically-correct ratio.
> 5. Optimize πSFT with CHRPO using Rmain for rationale execution and Rctrl for first-token ratio selection.
>
> Output: a controllable reasoning model supporting both fixed-ratio tokens and <COMP_POLICY>.
> ```
>
> ### Q4. Necessity and Motivation of Each RL Reward
>
> **Concern:** The rewards in the RL objective are not clearly defined or motivated.
>
> Training a policy to compress its own thoughts is highly unstable. Rather than ad-hoc terms, our RL components address specific failure modes to stabilize this process:
>
> - **Accuracy Reward ($R_{acc}$):** Optimizes final correctness.
> - **Ratio-Optimized Mode Reward ($R_{mode}$):** Encourages lower budgets **only when correct**. Penalizing failure from over-aggressive shortening teaches the model to use more tokens on hard problems.
> - **Budget Calibration Reward ($R_{cal}$):** Ensures actual generated reasoning length matches the promised budget.
> - **Rationale Integrity Reward ($R_{\emptyset}$):** Prevents the policy from bypassing the control mechanism by omitting `<think>...</think>` entirely.
> - **Control-Head Reward ($R_{ctrl}$):** Ratio selection occurs at **the first token**, but correctness is observed at the **end of the sequence**. $R_{ctrl}$ provides an immediate, localized reward to the selection token, preventing policy instability from delayed signals.
>
> ### Q5. Clarification of the "Realized Ratio"
>
> **Concern:** Can you expand on the definition of realized ratio?
>
> We apologize for the lack of clarity. There is a crucial distinction between the **Target Ratio** ($r_c$) and the **Realized Ratio** ($\hat{r}$):
>
> - **Target Ratio ($r_c$):** The budget the policy aims for (e.g., `<COMP_20>`).
> - **Realized Ratio ($\hat{r}$):** The actual compression ratio produced (length of the generated trace strictly inside `<think>...</think>` divided by the original uncompressed length).

---

> > ### Author Rebuttal · Reviewer_odPM · 2026-04-03
> >
> > Thank you to the authors for the detailed responses and clarifications. I appreciate the additional explanation and my concerns have been addressed.

---

> > > ### Author Response · Authors · 2026-04-05
> > >
> > > Dear Reviewer odPM,
> > >
> > > Thank you again for your invaluable time and effort on our paper. Thank you very much for approving our work and rebuttal!
> > >
> > > Sincerely yours,
> > >
> > > The Authors

---

### Decision · Program_Chairs · 2026-04-30

**Decision:**

Accept (regular)

**Comment:**

This paper proposes Extra-CoT, a three-stage framework for extreme-ratio CoT compression that achieves over 73% token reduction on MATH-500 while slightly improving accuracy. All four reviewers rated Weak Accept and three confirmed their concerns fully resolved after rebuttal.

Reviewers agree the empirical results are strong, the ablations are thorough, and code is released. The main limitation is that the compressor is math-specialized; however, the RL component (CHRPO) is domain-agnostic. One reviewer's residual concern about hyperparameter count was addressed with a simplified 2-group parameterization achieving near-identical results.

I find this to be a solid contribution to CoT compression with strong empirical support. I recommend acceptance.